# Bridging and Modeling Correlations in Pairwise Data for Direct Preference Optimization

**Yuxin Jiang**[1,2]***Bo Huang**[1,2], **Yufei Wang**[3], **Xingshan Zeng**[3], **Liangyou Li**[3],
**Yasheng Wang**[3], **Xin Jiang**[3], **Lifeng Shang**[3], **Ruiming Tang**[3], **Wei Wang**[1,2]
[1]The Hong Kong University of Science and Technology (Guangzhou)
[2]The Hong Kong University of Science and Technology
[3]Huawei Noah's Ark Lab
yjiangcm@connect.ust.hk, weiwcs@ust.hk

## ABSTRACT

Direct preference optimization (DPO), a widely adopted offline preference optimization algorithm, aims to align large language models (LLMs) with human-desired behaviors using pairwise preference data. However, the generation of the winning response and the losing response within pairwise data are typically isolated, leading to weak correlations between them as well as suboptimal alignment performance. To address this issue, we propose an effective framework for Bridging and Modeling Correlations in pairwise data, named **BMC**. Firstly, we increase the consistency and informativeness of the pairwise preference signals through *targeted modifications*, synthesizing a pseudo-winning response by improving the losing response with the winning response as a reference. Secondly, we identify that DPO alone is insufficient to model these correlations and capture nuanced variations. Therefore, we propose learning token-level correlations by *dynamically* leveraging the policy model's confidence during training. Comprehensive experiments on QA, math, and instruction-following tasks demonstrate the effectiveness of our approach, significantly surpassing competitive baselines, including DPO. Additionally, our in-depth quantitative analysis reveals the reasons behind our method's superior performance over DPO and showcases its versatility to other DPO variants. We release our repository at https://github.com/YJiangcm/BMC.

## 1 INTRODUCTION

Direct preference optimization (DPO) (Rafailov et al., 2023) has emerged as a prominent alternative to reinforcement learning from human feedback (RLHF) (Christiano et al., 2017; Bai et al., 2022a; Ouyang et al., 2022) for aligning large language models (LLMs) with human values. Unlike the traditional RLHF approach, DPO bypasses training a reward model and avoids using any reinforcement learning algorithms. Since the inception of DPO, numerous studies have sought to advance this method by refining its training objective (Wang et al., 2024). For instance, IPO (Azar et al., 2024) introduces an alternative pairwise preference loss to mitigate overfitting to the preference dataset, while R-DPO (Park et al., 2024) incorporates a regularization term to prevent the exploitation of latent length bias in the training data.

However, relatively little attention has been given to enhancing DPO through advancements in the quality of preference data used for training. In particular, the generation of winning and losing responses within preference data often occurs in an *isolated* manner, either through human annotation (Bai et al., 2022a) or automated techniques such as RLAIF (Bai et al., 2022b) and reject sampling (Liu et al., 2024a; Pace et al., 2024). This isolation implies that winning and losing responses are produced without mutual visibility, resulting in a lack of strong correlation or relevance between them. Consequently, the model may struggle to identify nuanced yet significant distinctions that differentiate superior responses from inferior ones (Fürnkranz & Hüllermeier, 2010; Wirth et al., 2017), which can ultimately compromise optimization and alignment effectiveness.

---

*Work partially done during the internship at Huawei Noah's Ark Lab.

In this work, we introduce an innovative framework, termed **BMC**, to Bridge and Model Correlations in pairwise data for direct preference optimization. During the Bridging Phase, we enhance correlations by increasing the consistency and informativeness of pairwise preference signals. By using the winning response as a reference, we synthesize a pseudo-winning response through *targeted modifications* of the losing response. This pseudo-winning response offers two key advantages: (1) it preserves essential characteristics of the losing response, minimizing noise in preference signals (*consistency*); (2) it encapsulates all human-desired values from the winning response, enabling the model to better discern features that lead to superior performance (*informativeness*). The nuanced differences between the pseudo-winning and losing responses are indeed what we expect the model to learn in the subsequent Modeling Phase. Nonetheless, we identify that DPO alone is insufficient to model these correlations and capture nuanced variations. From the perspective of the token-level Markov Decision Process (MDP) (Rafailov et al., 2024), DPO aggregates rewards uniformly across all tokens, assuming equal contribution to sequence quality and neglecting token-specific importance. To address this, we adjust the emphasis on rewards of different tokens between pseudo-winning and losing responses. Unlike previous methods (Guo et al., 2024; Cao et al., 2024; Chan et al., 2024; Chen et al., 2024a) that assign predefined values for fine-grained guidance, our adjustment is *dynamically* guided by the policy model's confidence, *i.e.*, the probability assigned to generated tokens during training. This ensures the model focuses on learning challenging distinctions while reinforcing known patterns, resulting in a more nuanced and robust policy.

We conduct extensive experiments across three downstream scenarios: question answering, mathematical reasoning, and instruction following, utilizing a total of 10 datasets. Our results demonstrate that our method consistently and significantly outperforms competitive offline optimization algorithms across various tasks. Furthermore, we use in-depth analyses to elucidate why our method outperforms DPO and show that our framework can be versatilely adapted to other DPO variants, confirming its potential for broad application.

## 2 RELATED WORK

**Preference optimization.** Preference optimization refers to aligning large language models with human preferences or specific desired outcomes. A well-established method for this is reinforcement learning from human feedback (RLHF) (Christiano et al., 2017; Bai et al., 2022a; Ouyang et al., 2022). Although RLHF produces highly capable models, its training process is often complex and unstable (Santacroce et al., 2023). To address these challenges, direct preference optimization (DPO) (Rafailov et al., 2023) introduces an alternative offline algorithm to optimize the regularized expected rewards without relying on RL. Building on DPO, subsequent methods like IPO(Azar et al., 2024) address overfitting to preference data, while R-DPO (Park et al., 2024) introduces length-based regularization to mitigate exploitation.

**Preference data construction.** Constructing high-quality pairwise preference data is essential for preference optimization. Given the high cost of manually curating these datasets at scale, researchers have explored automated methods for producing preference data. One notable approach, RLAIF (Bai et al., 2022b) employs LLMs to label side-by-side response pairs, eliminating the need for human labeling. Alternatively, winning and losing responses can be generated by utilizing models of varying quality (Kim et al., 2023) or through specific prompting techniques (Yang et al., 2023a). Recently, sampling-based methods such as Statistical Rejection Sampling (Liu et al., 2024a) and West-of-N (Pace et al., 2024) have been introduced, generating preference pairs by selecting candidates sampled from the optimal policy. Nonetheless, these methods isolatedly generate winning and losing responses without accounting for the correlations between them.

**Token-level preference optimization.** The majority of preference optimization strategies typically utilize trajectory-wise (sequence-level) rewards, while LM training and generation both occur at the token level (Yang et al., 2023b). To bridge this gap, FIGA (Guo et al., 2024) and DRLC (Cao et al., 2024) exploit external LLMs to pinpoint positive and negative token segments within responses, assigning fixed reward values (*e.g.*, +1 for positive, -1 for negative) as guidance. Meanwhile, ABC (Chan et al., 2024) and RLMEC (Chen et al., 2024a) extract fine-grained credits from the reward model. Despite their contributions, these methods rely on predefined values for fine-grained guidance, failing to account for the dynamic learning process of the policy model.

## 3   METHODOLOGY

In this section, we present the proposed **BMC** approach, which bridges and models correlations in pairwise data for direct preference optimization. As depicted in Figure 1, our BMC framework is structured around two pivotal stages: (1) the Bridging Phase, where we enhance the correlations between pairwise data by increasing the consistency and informativeness of pairwise preference signals through *targeted modifications* (§3.1); and (2) the Modeling Phase, where we *dynamically* model the correlations during the optimization process by leveraging the confidence of the policy model (§3.2), alleviating the insufficient token-level credit assignment of DPO.

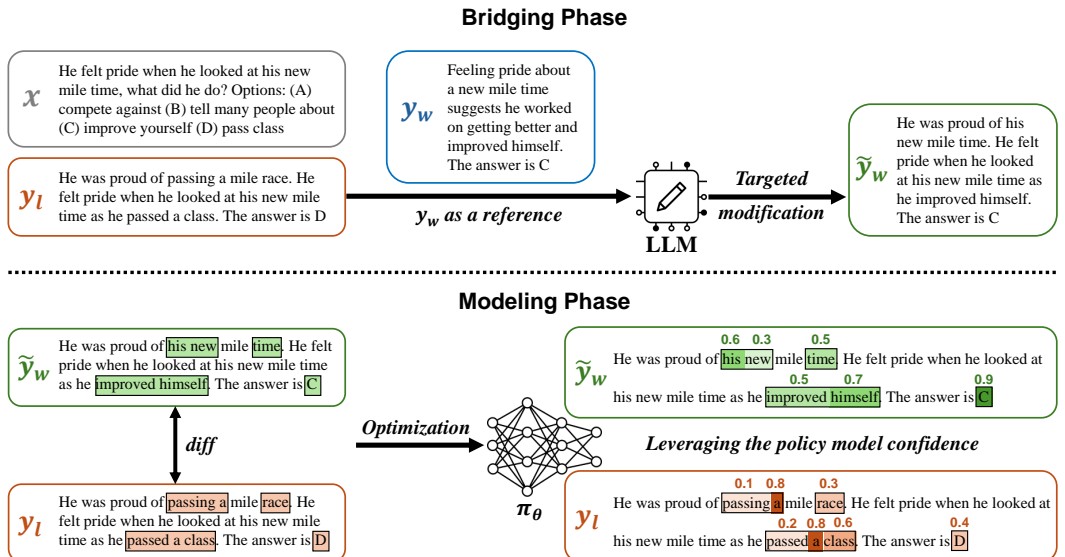

Figure 1: Overview of our proposed BMC framework. (1) In the Bridging Phase, we utilize an off-the-shelf LLM to make *targeted modifications* of losing response $y_l$ on undesired tokens, with the winning response $y_w$ serving as a reference. Therefore, the synthesized pseudo-winning response $\tilde{y}_w$ is highly correlated with $y_l$. (2) In the Modeling Phase, we model the correlations between $\tilde{y}_w$ and $y_l$ by *dynamically* emphasizing the rewards of their varied tokens ( $diff(\tilde{y}_w \mid y_l)$ and $diff(y_l \mid \tilde{y}_w)$), leveraging the policy model confidence (numbers indicated above tokens) during training.

### 3.1   BRIDGING PHASE

In offline preference optimization, it is commonly assumed that we have access to a static pairwise preference dataset $\mathcal{D} = \{x^{(i)}, y_w^{(i)}, y_l^{(i)}\}_{i=1}^N$, where $y_w$ and $y_l$ denote the winning and losing response, give the input prompt $x$. However, since $y_w$ and $y_l$ are typically generated in isolation, the correlation between $y_w$ and $y_l$ can be inherently weak during pairwise preference optimization. In the context of DPO, the Bradley-Terry objective (Bradley & Terry, 1952) computes gradients based on the relative likelihoods of $y_w$ and $y_l$. When the correlation between $y_w$ and $y_l$ is weak, the differences between these responses are often superficial (*e.g.*, stylistic or irrelevant variations) rather than substantive distinctions that reflect human-preferred behaviors. Consequently, the optimization process may inadvertently focus on minor discrepancies rather than meaningful distinctions. This results in gradients that are less informative for guiding the model towards robust preference alignment. To address this challenge, we enhance the alignment efficacy by improving the consistency and informativeness of pairwise preference signals. As shown in the upper part of Figure 1, we utilize an off-the-shelf LLM to make targeted modification of $y_l$ by referring to $y_w$:

$$\text{LLM}(I, x, y_w, y_l) \rightarrow \tilde{y}_w, \tag{1}$$

where $\tilde{y}_w$ is the generated pseudo-winning response, $I$ is the instruction (see examples in Appendix A.2) that requires $y_l$ to be modified only on dispreferred tokens, using $y_w$ as a reference guidance. In this way, $\tilde{y}_w$ preserves essential characteristics of the losing response $y_l$ while encapsulating all

human-desired values in the winning response $y_w$. The token-level differences between $\tilde{y}_w$ and $y_l$ highlight the core human expected and unexpected behaviors by decoupling from the inherent linguistic style and overall semantic distribution. Thus, $(\tilde{y}_w, y_l)$ refines the original training data $(y_w, y_l)$ for more focused learning, shifting the optimization process to concentrate on the most critical differences in preference data. The benefits of the Bridging Phase are further analyzed in §5.2. Finally, we use the new dataset $\tilde{\mathcal{D}} = \{x^{(i)}, \tilde{y}_w^{(i)}, y_l^{(i)}\}_{i=1}^N$ for subsequent training.

An alternative approach that attempts to enhance the correlation between the winning and losing responses is to degenerate $y_w$ to $\tilde{y}_l$ via targeted modification and utilize $(y_w, \tilde{y}_l)$ as the preference pair. Nevertheless, our ablation study in Table 3 reveals that LLMs encounter challenges with this inverse operation, leading to a notable decline in performance.

## 3.2 MODELING PHASE

After the Bridging Phase, the token-level differences between $\tilde{y}_w$ and $y_l$ can be obtained through dynamic programming algorithms like Levenshtein Distance (Yujian & Bo, 2007). As depicted in the lower part of Figure 1, these nuanced variations guide LLMs to prioritize the reinforcement of optimal actions while discouraging suboptimal ones within a single response. However, our findings below indicate that DPO alone is insufficient for capturing the nuanced variations, highlighting the necessity for supplementary techniques to comprehensively model these correlations.

**Alternative interpretation of DPO.** DPO (Rafailov et al., 2023) introduced a novel framework for optimizing the equivalent KL-constrained reward function as in RLHF, without the need to learn an explicit reward model. Instead, the problem is cast as a maximum likelihood estimation for the policy model $\pi_\theta$ on the preference dataset $\mathcal{D}$, resulting in the following training objective:

$$\mathcal{L}_{\text{DPO}}(\pi_\theta; \pi_{\text{ref}}) = -\mathbb{E}_{(x, y_w, y_l) \sim \mathcal{D}} \left[ \log \sigma \left( \beta \log \frac{\pi_\theta(y_w \mid x)}{\pi_{\text{ref}}(y_w \mid x)} - \beta \log \frac{\pi_\theta(y_l \mid x)}{\pi_{\text{ref}}(y_l \mid x)} \right) \right], \quad (2)$$

where $\pi_{\text{ref}}$ is the reference model, typically the supervised fine-tuned (SFT) model, and $\beta$ is a regularisation term corresponding to the strength of KL-regularization in RLHF.

As shown in Eq. (2), DPO was originally conceptualized as a bandit problem, where the whole response of the model is treated as a single arm to receive a reward. More recently, Rafailov et al. (2024) extended the theoretical foundation of DPO, showing that it can also be derived in the context of token-level MDP. The corresponding training objective at the token level is:

$$\mathcal{L}_{\text{DPO}}(\pi_\theta; \pi_{\text{ref}}) = -\mathbb{E}_{(\tau_w, \tau_l) \sim \mathcal{D}} \left[ \log \sigma \left( \beta \sum_{t=0}^{N-1} \log \frac{\pi_\theta(a_w^t \mid s_w^t)}{\pi_{\text{ref}}(a_w^t \mid s_w^t)} - \beta \sum_{t=0}^{M-1} \log \frac{\pi_\theta(a_l^t \mid s_l^t)}{\pi_{\text{ref}}(a_l^t \mid s_l^t)} \right) \right], \quad (3)$$

where $\tau_w$ and $\tau_l$ denote the win trajectory and the lose trajectory, respectively. $a$ indicates the action (current generated token), and $s$ signifies the state (all tokens generated so far).

**Our solution.** It can be inferred from Eq. (3) that DPO, redefined as a token-level MDP, assigns rewards to each token generation by $\beta \log \frac{\pi_\theta(a^t \mid s^t)}{\pi_{\text{ref}}(a^t \mid s^t)}$, and simply add up the rewards of all tokens as the accumulated reward of the trajectory. This *uniform aggregation* assumes that each token contributes equally to the overall quality of the sequence, without considering the varying importance of each token (timestep). Therefore, nuanced differences between $\tilde{y}_w$ and $y_l$ that significantly influence the overall meaning or quality of the response might not be adequately emphasized (refer to Figure 6), leading to suboptimal performance. To this end, we propose to emphasize the rewards of critical tokens, *i.e.*, nuanced differences between $\tilde{y}_w$ and $y_l$. The magnitude of the emphasis is determined *dynamically* by the policy model's confidence, which refers to the probability assigned to the generated token during training. Below, we detail our design choices for the pseudo-winning response and losing response, respectively.

- For varied tokens in the pseudo-winning response $\tilde{y}_w$, we adapt the reward factor based on the learning process of the policy model. Lower policy confidence indicates underdeveloped learning of the target behavior, signaling the need for additional focus to help the model better capture these nuances. Consequently, we adjust the reward factor to be inversely proportional to the policy model's confidence, as formalized in Eq. (5).

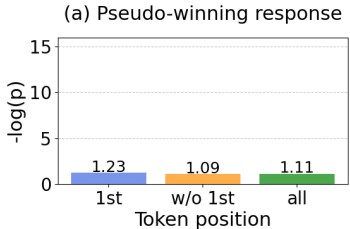 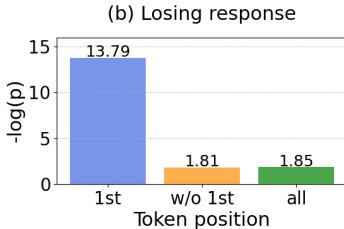

Figure 2: We aggregate varied tokens in $\tilde{y}_w$ or $y_l$ into more coarser-grained spans. During the DPO training on $\tilde{\mathcal{D}}$, we compute the averaged $-\log(p)$ of tokens in different positions of spans.

- For varied tokens in the losing response $y_l$, we carefully adjust the reward factor by reinforcing already learned patterns of the policy model. Intuitively, tokens in $y_l$ with higher confidence from the policy model may reflect inaccurate preference learning and therefore warrant stronger penalization. However, our analysis reveals a distinct pattern of the policy model when processing $y_l$ compared to $\tilde{y}_w$. Specifically, when grouping varied tokens in $y_l$ into coarser-grained spans, the model's confidence is significantly influenced by the token's position within these spans, as illustrated in Figure 2. We observe that the probabilities assigned to the initial token of incorrect spans in $y_l$ are typically low, whereas the probabilities for subsequent tokens within the same span are notably higher. Prior studies have identified token probability as a critical signal for detecting anomalous behaviors (Xiao & Wang, 2021; Fadeeva et al., 2024) and assessing generation quality (Yuan et al., 2021; Fu et al., 2024). Consistent with these findings, our results indicate that during training, the policy model can effectively recognize the onset of undesired spans by assigning low probabilities to initial tokens. Nonetheless, due to the autoregressive dependencies, subsequent tokens within these spans receive higher probabilities, reflecting the contextual coherence established by preceding tokens, even when the span as a whole is incorrect. Thus, while it is crucial to penalize initial tokens, applying equally strong penalties to subsequent tokens might be suboptimal, as they often maintain local coherence within the flawed span. Therefore, we adjust the reward factor to also be inversely proportional to the policy model's confidence in Eq. (6).

In a nutshell, our approach dynamically modulates the emphasis placed on critical tokens based on the policy model's confidence. This adaptive reward mechanism ensures that the model focuses on learning challenging distinctions while reinforcing already learned patterns, ultimately fostering a more nuanced and robust policy (see our analysis in §5.2). The formalization of our approach is encapsulated in Eq. (4), where $\lambda_{\tilde{y}_w^t}$ and $\lambda_{y_l^t}$ adjust dynamically based on the policy's confidence, ensuring a tailored emphasis on critical tokens to improve the overall model performance.

$$\mathcal{L}_{\text{DPO-BMC}}(\pi_\theta; \pi_{\text{ref}}) = -\mathbb{E}_{(x, \tilde{y}_w, y_l) \sim \tilde{\mathcal{D}}} \left[ \log \sigma \left( \beta \sum_{\tilde{y}_w^t \in \tilde{y}_w} \lambda_{\tilde{y}_w^t} \log \frac{\pi_\theta(\tilde{y}_w^t \mid \tilde{y}_w^{<t}, x)}{\pi_{\text{ref}}(\tilde{y}_w^t \mid \tilde{y}_w^{<t}, x)} \right. \right.$$
$$\left. \left. - \beta \sum_{y_l^t \in y_l} \lambda_{y_l^t} \log \frac{\pi_\theta(y_l^t \mid y_l^{<t}, x)}{\pi_{\text{ref}}(y_l^t \mid y_l^{<t}, x)} \right) \right], \qquad (4)$$

where

$$\lambda_{\tilde{y}_w^t} = \begin{cases} 1 + \min\left(sg\left(\frac{1}{\pi_\theta(\tilde{y}_w^t \mid \tilde{y}_w^{<t}, x)}\right), \delta\right), & \text{if } \tilde{y}_w^t \in diff(\tilde{y}_w \mid y_l) \\ 1, & \text{otherwise} \end{cases} \qquad (5)$$

$$\lambda_{y_l^t} = \begin{cases} 1 + \min\left(sg\left(\frac{1}{\pi_\theta(y_l^t \mid y_l^{<t}, x)}\right), \delta\right), & \text{if } y_l^t \in diff(y_l \mid \tilde{y}_w) \\ 1, & \text{otherwise} \end{cases} \qquad (6)$$

The $sg$ denotes the stop-gradient operator, the $\delta$ is an upper limit threshold that controls the emphasis on the rewards of the critical tokens, preventing overly aggressive updates. The $diff(\tilde{y}_w \mid y_l)$ and $diff(y_l \mid \tilde{y}_w)$ signify using the Levenshtein Distance algorithm to find the varied tokens in $\tilde{y}_w$ and $y_l$, respectively. In Appendix C, we provide a gradient analysis of DPO-BMC. Unlike DPO, **our approach harmonizes both sequential and token-level perspectives**, effectively optimizing the overall sequence structure alongside crucial token choices for the desired outcome.

## 4 EXPERIMENTAL SETUP

We conduct a comprehensive evaluation across three downstream scenarios, including question answering (QA), mathematical reasoning, and instruction following (IF). The detailed data statistics as well as the evaluation metrics are listed in Table 6 of Appendix A.1.

**Models and training settings.**    For the QA and mathematical reasoning setup, we utilize Llama2-7B-base (Touvron et al., 2023) in our experiments. Dealing with these tasks necessitates LLMs to possess domain-specific knowledge and engage in systematic, step-by-step reasoning to reach the ultimate answer. Therefore, following prior works (Chen et al., 2024a;b), we fine-tune Llama2-7B-base on the training set of ECQA (Aggarwal et al., 2021) and QASC (Khot et al., 2020) for QA, and fine-tune Llama2-7B-base on MetaMathQA (Yu et al., 2024) for mathematical reasoning. We denote the fine-tuned LLM as SFT and use it as the backbone for preference optimization. In line with prior research (Chen et al., 2024a;b), we construct preference pairs $(y_w, y_l)$ based on the training data, by using the ground truth as $y_w$ and the SFT model's inference output as $y_l$. For the instruction following setup, we utilize Llama3-8B-base (Dubey et al., 2024) and Mistral-7B-Base (Jiang et al., 2023a) in our experiments. Following the training pipeline of Zephyr (Tunstall et al., 2023) and SimPO (Meng et al., 2024), we train a base model on the UltraChat-200k dataset (Ding et al., 2023) to obtain an SFT model. Then, we use the SFT model as the starting point and perform preference optimization on the UltraFeedback dataset (Cui et al., 2023), where $y_w$ and $y_l$ are collected from LLMs of varying quality.

During our Bridging Phase, we utilize `gpt-4-0125-preview` for targeted modification to obtain $\tilde{y}_w$, based on the prompt template in Appendix A.2. We also demonstrate in Table 4 that **a less powerful open-source LLM, such as `Llama3-70B-Instruct`, can acquire comparable results**. During our Modeling Phase, we list the implementation details in Appendix A.3 for reproducibility. A comprehensive cost analysis in Appendix B confirms that **the computational overhead introduced by our BMC pipeline is minimal**.

**Evaluation benchmarks.**    In question answering, we adopt the test splits of ECQA (Aggarwal et al., 2021), QASC (Khot et al., 2020), OpenbookQA (Mihaylov et al., 2018), and StrategyQA (Geva et al., 2021) for evaluation. In mathematical reasoning, we conduct the evaluation on four challenge datasets including GSM8k (Cobbe et al., 2021), MATH (Hendrycks et al., 2021), MAWPS (Koncel-Kedziorski et al., 2016), and TabMWP (Lu et al., 2023). In instruction following, We assess our models using two of the most popular open-ended instruction-following benchmarks: AlpacaEval 2 (Li et al., 2023) and Arena-Hard v0.1 (Li et al., 2024). Both benchmarks evaluate the models' versatile conversational abilities across a diverse set of queries. For each query, the evaluated model's response and the reference model's response are compared head-to-head using an auto-evaluator. We use the officially recommended configurations[1] during the evaluation.

**Baselines.**    We compare our approach with various powerful *offline* preference optimization methods, including FIGA (Guo et al., 2024), DPO (Rafailov et al., 2023), and DPO variants (IPO (Azar et al., 2024), ORPO (Hong et al., 2024), R-DPO (Park et al., 2024), and SimPO (Meng et al., 2024)). The training objectives of these methods are listed in Table 7. Besides, we include two additional baselines: (1) **DPO (CW)**: enhancing pairwise data correlation by prompting the SFT model to **C**ontinue **W**riting a prefix of the winning response to generate the losing one; (2) **DPO (EW)**: leveraging an off-the-shelf LLM for **E**xternal **W**eighting of token-level reward (Lee et al., 2024), where LLM scores each token in the winning and losing responses based on how much it improves or decreases the overall quality.

## 5 EXPERIMENTAL RESULTS

In this section, we present the main results of our experiments, showcasing the superior performance of our method across various benchmarks and ablation studies (§5.1). Next, we conduct in-depth quantitative analyses to elucidate why our method outperforms DPO (§5.2 and §5.3). Furthermore, we demonstrate the versatility of our framework by adapting it to other DPO variants (§5.4).

---

[1]AlpacaEval: https://github.com/tatsu-lab/alpaca_eval. Arena-Hard v0.1: https://github.com/lm-sys/arena-hard-auto.

Table 1: Experimental results (based on Llama2-7B-base) on question answering tasks and mathematical reasoning tasks. "Avg." is the average accuracy of all sub-tasks. In each column, the highest score is **bolded** and the second-highest is underlined.

| Method | Question-Answering Tasks | | | | | Mathematical Reasoning Tasks | | | | |
|---|---|---|---|---|---|---|---|---|---|---|
| | ECQA | QASC | OBQA | StrategyQA | Avg. | GSM8k | MATH | MAWPS | TabMWP | Avg. |
| SFT | 72.8 | 54.5 | 51.8 | 56.9 | 59.0 | 55.8 | 11.6 | 80.3 | 42.8 | 47.6 |
| FIGA | 70.3 | 52.5 | 51.7 | 48.6 | 55.8 | 54.1 | 9.8 | 75.5 | 39.0 | 44.6 |
| IPO | 71.5 | 58.9 | 53.6 | 58.4 | 60.6 | 57.2 | 12.1 | 82.2 | 42.5 | 48.5 |
| OPRO | 69.8 | 55.1 | 51.4 | 57.2 | 58.4 | 56.0 | 12.4 | 80.8 | 41.3 | 47.6 |
| R-DPO | 73.5 | 59.5 | 55.4 | 58.8 | 61.8 | 56.9 | 12.0 | 81.9 | 42.2 | 48.2 |
| SimPO | 71.9 | 56.7 | 52.2 | 55.4 | 59.1 | 57.5 | 12.7 | 81.8 | 43.5 | 48.9 |
| DPO | 73.1 | 58.8 | 55.6 | 57.8 | 61.3 | 56.3 | 12.3 | 81.2 | 43.4 | 48.3 |
| DPO (CW) | 72.5 | 58.6 | 55.2 | 57.3 | 60.9 | 55.9 | 11.8 | 80.7 | 42.8 | 47.8 |
| DPO (EW) | 72.9 | 59.4 | 55.8 | 57.9 | 61.5 | 56.5 | 12.0 | 80.9 | 43.4 | 48.2 |
| DPO-BMC | **75.9** | **63.0** | **60.4** | **61.0** | **65.1** | **58.4** | **13.0** | **83.1** | **43.8** | **49.6** |
| DPO-BC | 75.7 | 62.0 | 56.0 | 60.1 | 63.4 | 57.6 | 12.7 | 82.8 | 43.4 | 49.1 |
| DPO-MC | 74.8 | 60.0 | 56.4 | 58.8 | 62.5 | 57.2 | 12.5 | 82.4 | 43.0 | 48.8 |

Table 2: Experimental results on instruction-following tasks. "LC" is the length-controlled win rate, and "WR" is the raw win rate. "Avg. len" denotes the average number of tokens in the responses.

| Method | Llama3-8B-Base | | | | | Mistral-7B-Base | | | | |
|---|---|---|---|---|---|---|---|---|---|---|
| | AlpacaEval 2 | | | Arena-Hard | | AlpacaEval 2 | | | Arena-Hard | |
| | LC (%) | WR (%) | Avg. len | WR (%) | Avg. len | LC (%) | WR (%) | Avg. len | WR (%) | Avg. len |
| SFT | 7.5 | 4.7 | 956 | 2.6 | 414 | 8.1 | 5.9 | 998 | 2.2 | 454 |
| FIGA | 8.4 | 4.2 | 1,199 | 5.1 | 416 | 7.0 | 4.9 | 1,378 | 2.5 | 461 |
| IPO | 13.4 | 9.8 | 1,430 | 14.0 | 477 | 12.5 | 10.8 | 1,588 | 8.5 | 522 |
| ORPO | 12.5 | 11.4 | 1,793 | 11.7 | 573 | 14.5 | 11.5 | 1,630 | 9.4 | 566 |
| R-DPO | 17.1 | 14.4 | 1,801 | 17.6 | 582 | 16.0 | 12.3 | 1,521 | 10.4 | 529 |
| SimPO | 21.3 | **18.9** | 1,718 | **26.6** | 562 | 16.8 | 14.4 | 1,906 | **18.4** | 615 |
| DPO | 16.0 | 14.8 | 1,713 | 17.6 | 559 | 15.1 | 13.3 | 1,657 | 13.6 | 540 |
| DPO (CW) | 15.2 | 14.0 | 1,756 | 17.1 | 570 | 14.5 | 12.9 | 1,647 | 13.0 | 532 |
| DPO (EW) | 17.2 | 15.6 | 1,702 | 18.2 | 566 | 15.3 | 13.4 | 1,668 | 13.9 | 549 |
| DPO-BMC | **22.4** | 16.8 | 1,285 | 18.1 | 406 | **20.8** | **16.6** | 1,317 | 17.6 | 488 |
| DPO-BC | 20.6 | 14.4 | 1,269 | 16.8 | 422 | 18.6 | 13.8 | 1,489 | 15.9 | 502 |
| DPO-MC | 17.7 | 15.2 | 1,890 | 17.9 | 579 | 16.4 | 14.3 | 1,712 | 15.4 | 551 |

## 5.1 MAIN RESULTS AND ABLATIONS

**Our method consistently and significantly outperforms baselines.** As presented in Table 1, our model DPO-BMC consistently achieves state-of-the-art results across all evaluated QA and math benchmarks. Specifically, DPO-BMC outperforms DPO by 3.8 absolute points on QA tasks and by 1.3 points on math tasks. On instruction-following tasks (Table 2), DPO-BMC secures the highest length-controlled win rate, surpassing DPO by over 5 points across various settings, with even greater gains for larger base models (Appendix E). The length-controlled win rate (Dubois et al., 2024) serves as a robust metric that mitigates the effects of length bias, thereby providing a more reliable evaluation of LLM-based auto-annotation. Notably, **DPO-BMC generates responses that are significantly more concise than other baselines**. As highlighted in Table 2, the average response length of DPO-BMC and DPO-BC is approximately 75% of that produced by DPO and DPO-MC. This attribute of length normalization is credited to the correlated preference data we constructed, which directs optimization towards critical desired behaviors rather than verbosity. A case study in Table 12 further underscores the effectiveness and robustness of our approach.

**Both key designs in BMC are crucial.** In Table 1 and Table 2, we additionally present results from ablating each key design element of DPO-BMC:

• **DPO-BC**: Training using DPO's original objective on our constructed preference data.

• **DPO-MC**: Training using our proposed objective in Eq. (4) on the original preference data.

Table 3: Ablation study on diverse data synthesis methods in the Bridging Phase. The average accuracy is presented for QA and Math. LC on AlpacaEval 2 is reported for instruction following (IF), based on Llama3-8B.

| Data Synthesis | Training Data | QA | Math | IF |
|---|---|---|---|---|
| $y_l \xrightarrow{y_w} \tilde{y}_w$ (ours) | $(\tilde{y}_w, y_l)$ | **65.1** | **49.6** | **22.4** |
| $y_l \longrightarrow \tilde{y}_w$ | $(\tilde{y}_w, y_l)$ | 64.3 | 49.2 | 19.8 |
| $y_w \xrightarrow{y_l} \tilde{y}_l$ | $(y_w, \tilde{y}_l)$ | 64.6 | 48.7 | 18.9 |
| $y_w \longrightarrow \tilde{y}_l$ | $(y_w, \tilde{y}_l)$ | 63.9 | 48.6 | 17.6 |

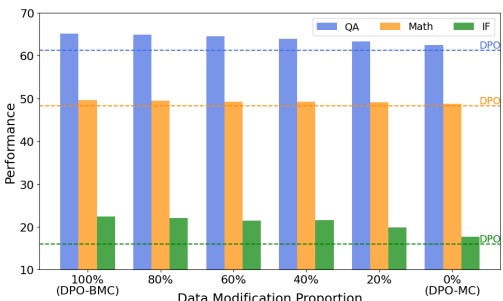

Figure 3: Ablation study on data modification proportion in the Bridging Phase.

Our examination reveals several key findings: (1) DPO (CW), the "Continue Writing" approach, slightly underperforms standard DPO, as it introduces superficial correlations that fail to capture the nuanced, task-specific alignments essential for effective optimization. In contrast, our Bridging Phase explicitly enhances *informative correlations*—elucidate fine-grained distinctions between desired and undesired behaviors through token-level variations. This targeted focus significantly improves model performance; (2) Even when leveraging identical training preference data, our designed optimization objective consistently outperforms both DPO and DPO (EW), highlighting its superior ability to model fine-grained correlations based on the dynamic of the policy model's confidence; and (3) Combining our constructed data with our designed objective yields the best results, affirming the inseparability of the Bridging Phase and the Modeling Phase.

**Influence of data synthesis method.** Table 3 shows the effects of various data synthesis strategies during the Bridging Phase. When generating $\tilde{y}_w$ without referring to $y_w$, LLMs potentially make erroneous modifications that misalign with the intended target, leading to a performance drop. An alternative approach that attempts to enhance the correlation between winning and losing responses is to degenerate $y_w$ to $\tilde{y}_l$ and utilize $(y_w, \tilde{y}_l)$ as the preference pair. However, this approach also falls short, likely because LLMs are primarily trained to generate high-quality data, making it challenging for them to generate low-quality outputs that mimic the nuanced errors of losing responses. Semantic similarity analysis using the `all-mpnet-base-v2` embedding model[2] supports this, showing a high score of 0.88 for $(y_w, \tilde{y}_w)$ but only 0.73 for $(y_l, \tilde{y}_l)$.

**Influence of data modification proportion.** Figure 3 illustrates the impact of data modification proportions during the Bridging Phase on performance. Increasing modifications from 0% to 20% yields the most substantial gains, highlighting the effectiveness of enhancing pairwise preference correlations. Performance plateaus beyond 80% modifications, indicating that extensive changes are beneficial but not essential, offering flexibility under computational or data constraints. These results demonstrate the scalability and adaptability of our framework for diverse applications.

**Influence of LLMs for targeted modification.** Table 4 explores the influence of diverse LLMs for targeted modification. Notably, substituting the `gpt-4-0125-preview` model with a less powerful yet open-source alternative, such as `Llama3-70B-Instruct`, **yields comparable performance while significantly surpassing vanilla DPO**. This finding underscores the adaptability of our method to varying levels of model sophistication, thereby reducing dependence on commercial LLMs without significant impact on final model performance.

**Influence of $\delta$.** We conduct an ablation study to examine the influence of the threshold $\delta$ in the DPO-BMC objective on model performance, as shown in Figure 4. Setting $\delta = 1.0$ reduces our method to one that assigns fixed token-level rewards, leading to suboptimal accuracy. As $\delta$ increases, the model performance improves, with the optimal setting observed around $\delta = 3.0$. However, further increasing $\delta$ results may degrade model performance due to excessively aggressive gradient updates on certain tokens. Notably, across all tested values of $\delta$, our method consistently outperforms the DPO baseline, indicating its robustness and effectiveness in stabilizing the learning process.

---

[2]https://huggingface.co/sentence-transformers/all-mpnet-base-v2

Table 4: Influence of diverse LLMs for targeted modification in the Bridging Phase. The average accuracy is presented for QA and Math. LC on AlpacaEval 2 is reported for instruction following (IF), based on Llama3-8B.

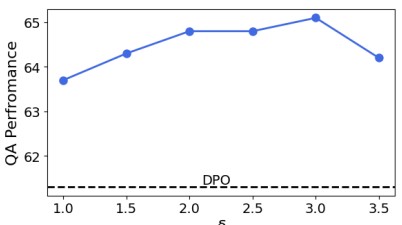

| Method | LLM for Targeted Modification | QA | Math | IF |
|--------|-------------------------------|------|------|------|
| SFT | – | 56.9 | 47.6 | 7.5 |
| DPO | – | 61.3 | 48.3 | 16.0 |
| DPO-BMC | Llama3-70B-Instruct | 64.6 | 49.4 | 21.8 |
| DPO-BMC | gpt-4-0125-preview | **65.1** | **49.6** | **22.4** |

Figure 4: Ablation study on $\delta$ in the Modeling Phase. The average accuracy is presented as the QA performance.

**(a) DPO**

| $(y_w, y_l)$ | Edit Distance | LC (%) | Grad Norm |
|--------------|---------------|--------|-----------|
| split 1 | 0.57 | 7.68 | 3.31 |
| split 2 | 0.70 | 9.49 | 4.85 |
| split 3 | 0.73 | 10.50 | 4.86 |
| split 4 | 0.76 | 10.01 | 5.33 |
| split 5 | 0.83 | 8.57 | 6.31 |
| split 6 | 0.95 | 7.91 | 13.00 |

**(b) DPO-MC**

| $(y_w, y_l)$ | Edit Distance | LC (%) | Grad Norm |
|--------------|---------------|--------|-----------|
| split 1 | 0.57 | 9.40 | 5.70 |
| split 2 | 0.70 | 12.49 | 8.39 |
| split 3 | 0.73 | 13.27 | 8.66 |
| split 4 | 0.76 | 11.47 | 9.03 |
| split 5 | 0.83 | 9.81 | 8.44 |
| split 6 | 0.95 | 9.90 | 9.04 |

**(c) DPO-BC**

| $(\tilde{y}_w, y_l)$ | Edit Distance | LC (%) | Grad Norm |
|--------------|---------------|--------|-----------|
| split 1 | 0.45 | 10.82 | 3.47 |
| split 2 | 0.52 | 10.87 | 4.80 |
| split 3 | 0.56 | 12.54 | 5.20 |
| split 4 | 0.61 | 14.34 | 5.39 |
| split 5 | 0.70 | 13.24 | 6.98 |
| split 6 | 0.84 | 10.59 | 9.67 |

**(d) DPO-BMC**

| $(\tilde{y}_w, y_l)$ | Edit Distance | LC (%) | Grad Norm |
|--------------|---------------|--------|-----------|
| split 1 | 0.45 | 11.21 | 5.26 |
| split 2 | 0.52 | 11.49 | 7.33 |
| split 3 | 0.56 | 11.47 | 7.70 |
| split 4 | 0.61 | 14.38 | 8.17 |
| split 5 | 0.70 | 15.28 | 7.65 |
| split 6 | 0.84 | 12.29 | 8.75 |

Figure 5: We segment the 60k training data of UltraFeedback into six equal-sized splits based on increasing edit distance between winning and losing responses. For each split, we report LC on AlpacaEval 2 and the average gradient norm during training.

## 5.2 QUANTITATIVE ANALYSIS OF BRIDGING AND MODELING PHASE

To rigorously assess the effectiveness of the two pivotal phases in our framework, we segment the 60k training data of UltraFeedback into six equal-sized splits, ordered by increasing edit distance between winning and losing responses. For each split, we also construct its corresponding $(\tilde{y}_w, y_l)$ pair data through our Bridging Phase. We then train four models——(a) DPO, (b) DPO-MC, (c) DPO-BC, and (d) DPO-BMC—on each split based on Llama3-8B, with identical hyperparameters to ensure comparability. As shown in Figure 5, the Bridging Phase successfully decreases the edit distance between pairwise data through targeted modification, shifting the optimization process to concentrate on the most critical differences in preference data. This phase consistently enhances performance across all splits by refining training data for more focused learning. Another notable observation is the average gradient norm during DPO training increases as the edit distance between pairwise data enlarges, reflecting the sensitivity of DPO's training process to individual data points and potential gradient variance. Our proposed Modeling Phase mitigates the variance by dynamically adjusting the training process based on the policy model's confidence. This adaptive mechanism prioritizes challenging distinctions while reinforcing learned patterns, promoting a balanced optimization landscape with diverse training data (See Appendix D for further analysis).

## 5.3 QUANTITATIVE ANALYSIS OF CREDIT ASSIGNMENT

We compare the token-level and sequence-level credits assigned by DPO and DPO-BMC, assessing how well their final learned rewards align with preference labels on a held-out set of UltraFeedback.

**Analysis on token-level reward.** Figure 6 depicts the token-level reward assignment for DPO and DPO-BMC on a response pair consisting of a winning response $y_w$ and a losing response $y_l$. The reward of each token is computed as $r_\theta(x, y^t) = \beta \log \frac{\pi_\theta(y^t|y^{<t},x)}{\pi_{\text{ref}}(y^t|y^{<t},x)}$. From the figure, we observe that: (1) DPO assigns nearly uniform rewards across tokens, failing to differentiate the importance of tokens to the overall response quality; and (2) although DPO can identify and assign lower rewards to several erroneous tokens in the losing response (*e.g.*, "13"), it struggles to capture

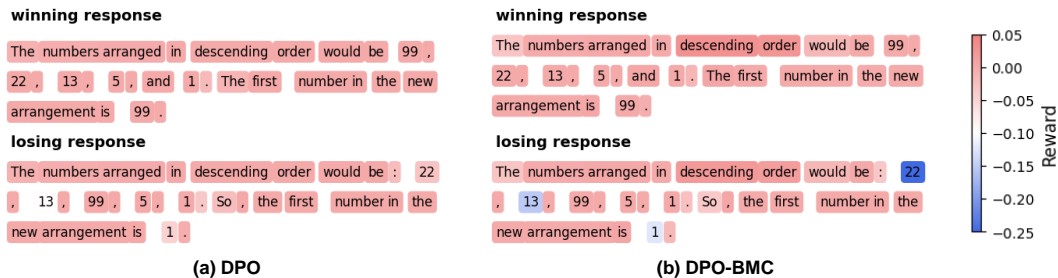

Figure 6: Visualization of token-level rewards assigned by DPO and our method. The preference pair is sampled from the held-out set of UltraFeedback, whose input prompt is "*Arrange the numbers 5, 13, 99, 1, and 22 in descending order. What is the first number in the new arrangement?*"

subtle distinctions between the winning and losing responses. In contrast, DPO-BMC assigns higher rewards to critical tokens (*e.g.*, "descending order") and effectively penalizes incorrect tokens in the losing response. These results demonstrate DPO's limitations in providing precise token-level preferences on sentence quality, and our method can effectively alleviate this issue.

**Analysis on sequence-level reward.** For a rigorous comparison, we calculate the sequence-level DPO reward expression by $r_\theta(x, y) = \beta \log \frac{\pi_\theta(y|x)}{\pi_{\text{ref}}(y|x)}$. The reward margin is determined by $r_\theta(x, y_w) - r_\theta(x, y_l)$. Reward accuracy is defined as the percentage of preference pairs where the winning response achieves a higher reward than the losing response, *i.e.*, $r_\theta(x, y_w) > r_\theta(x, y_l)$. Our findings show that DPO-BMC outperforms DPO in terms of average reward margin (**0.74 vs. 0.54**) and reward accuracy (**73.60 vs. 72.19**). This enhancement validates our method's superior ability to discern subtle differences between preference pairs, enabling more effective generalization.

## 5.4 VERSATILITY OF OUR FRAMEWORK

Our BMC framework demonstrates versatility and can be seamlessly integrated with various DPO variants. As shown in Table 5, the XPO-BMC methods consistently outperform their corresponding XPO baselines across a diverse set of tasks, including QA, Math, and Instruction Following (IF). For instance, IPO-BMC achieves a significant improvement in QA accuracy (64.1 vs. 60.6) and IF score (15.7 vs. 13.4) compared to IPO. Similarly, ORPO-BMC, R-DPO-BMC, SimPO-BMC, and DPO-BMC exhibit higher performance in QA and Math, alongside notable gains in IF, such as R-DPO-BMC improving the IF score from 17.1 to 20.0 over R-DPO. These results highlight the robustness of our framework in enhancing task-specific performance across various settings, reaffirming its potential as a generalizable enhancement to existing DPO methodologies.

Table 5: Versatility of our framework across various XPOs..

| Method | QA | Math | IF |
|---|---|---|---|
| SFT | 56.9 | 47.6 | 7.5 |
| IPO | 60.6 | 48.3 | 13.4 |
| IPO-BMC | **64.1** | **48.6** | **15.7** |
| ORPO | 58.4 | 47.6 | 12.5 |
| ORPO-BMC | **62.3** | **48.4** | **15.7** |
| R-DPO | 61.8 | 48.2 | 17.1 |
| R-DPO-BMC | **65.3** | **49.5** | **20.0** |
| SimPO | 59.1 | 48.9 | 21.3 |
| SimPO-BMC | **61.6** | **49.0** | **21.9** |
| DPO | 61.3 | 48.3 | 16.0 |
| DPO-BMC | **65.1** | **49.6** | **22.4** |

## 6 CONCLUSION

In this work, we propose BMC, an effective framework for bridging and modeling correlations in pairwise data for direct preference optimization. BMC equips LLMs with better human value alignment through a two-phase process: a Bridging Phase that enhances correlations between pairwise data by explicitly manifesting fine-grained preference signals via targeted modifications, and a Modeling Phase that learns token-level correlations by dynamically leveraging the the policy model's confidence during training. Our framework exhibits superior performance in question-answering, mathematical reasoning, and instruction-following tasks, consistently surpassing the baseline DPO by a significant margin. Extensive analysis highlights that the key designs in BMC are crucial and validates the effectiveness and versatility of BMC.

ACKNOWLEDGMENTS

Wei Wang was supported by the Guangdong Provincial Key Laboratory of Integrated Communication, Sensing, and Computation for Ubiquitous Internet of Things (Grant No. 2023B1212010007), the Guangzhou Municipal Science and Technology Project (Grant Nos. 2023A03J0003, 2023A03J0013, and 2024A03J0621), and the Institute of Education Innovation and Practice Project (Grant Nos. G01RF000012 and G01RF000017).

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

## APPENDICES

## A DETAILED EXPERIMENTAL SETUP

### A.1 DATA STATISTICS AND EVALUATION METRICS USED FOR EXPERIMENTS

We list the detailed data statistics and evaluation metrics of our experiments in Table 6. Our experiments comprise both closed-ended evaluation (QA and math) and open-ended evaluation (instruction following).

Table 6: Statistics of the training and evaluation datasets.

| Task | Train / Test | Dataset | Number | Evaluation Metric |
|------|------|------|------|------|
| QA | Train | ECQA | 7,598 | |
| | | QASC | 8,134 | |
| | Test | ECQA | 2,194 | Accuracy |
| | | QASC | 926 | |
| | | OBQA | 500 | |
| | | StrategyQA | 687 | |
| Math | Train | MetaMathQA | 40,000 | |
| | Test | GSM8k | 1,319 | Accuracy |
| | | MATH | 5,000 | |
| | | MAWPS | 2,065 | |
| | | TabMWP | 1,000 | |
| IF | Train | UltraFeedback | 61,135 | |
| | Test | AlpacaEval 2 | 805 | Win rate against GPT-4 Turbo |
| | | Arena-Hard | 500 | |

### A.2 PROMPT TEMPLATE FOR TARGETED MODIFICATION

We demonstrate the prompt template of targeted modification for question answering and mathematical reasoning tasks in Figure 7. Since the SFT model has been fine-tuned on the ground truth $y_w$ for QA and math tasks, the inferred output $y_l$ may be quite approximate to $y_w$ in some circumstances. Therefore, we require the off-the-shelf LLM to filter out preference pairs where $y_l$ is good enough. Finally, we filtered out 31% data and 43% data for the QA task and math task, respectively. **Note that for the training data of our baselines like DPO, we also use the filtered $(y_w, y_l)$ pairs for a fair comparison.** For instruction-following tasks, the prompt template we use is shown in Figure 8.

### A.3 IMPLEMENTATION DETAILS

Our implementation is based on the alignment-handbook repo[3] using 4×A800 GPUs. To ensure a fair comparison, we conduct thorough hyperparameter tuning for all methods compared in our experiments.

**SFT training hyperparameters.** We train SFT models using the following hyperparameters: a learning rate of 2e-5, a batch size of 128, a max sequence length of 2048, and a cosine learning rate schedule with 10% warmup steps. For QA and instruction-following tasks, we train the model for 1 epoch, whereas for mathematical tasks, we extend the training to 2 epochs. All the models are trained with an Adam optimizer (Kingma & Ba, 2014).

**Preference optimization training hyperparameters.** During preference optimization, we performed initial experiments to determine the optimal batch sizes in [32, 64, 128] and training epochs

---

[3]https://github.com/huggingface/alignment-handbook

---

**Prompt Template for Question Answering and Mathematical Reasoning Tasks**

**\*\*Task:\*\***
Given the Problem, the Correct Answer, and the Prediction, identify and correct any mistakes in the Prediction to align the Correct Answer. Three rules you must obey:
1. Make the minimal modifications necessary (changing the fewest words) to correct the Prediction.
2. Only output the complete Corrected Prediction without saying anything else.
3. If the Prediction is already good enough, simply output 'None'.

**\*\*Problem:\*\***
$\{x\}$

**\*\*Correct Answer:\*\***
$\{y_w\}$

**\*\*Prediction:\*\***
$\{y_l\}$

**\*\*Corrected Prediction:\*\***

---

Figure 7: Prompt template of targeted modification for question answering and mathematical reasoning tasks.

---

**Prompt Template for Instruction Following Tasks**

Task:\*\*
Below is a question followed by two responses. Response 1 is more preferred by humans than Response 2. Please make the minimal necessary changes to Response 2 to improve it, referring to Response 1. Maintain as much of the correct parts of Response 2 as possible. Output only the complete Revised Response 2.

**\*\*Problem:\*\***
$\{x\}$

**\*\*Response 1:\*\***
$\{y_w\}$

**\*\*Response 2:\*\***
$\{y_l\}$

**\*\*Revised Response 2:\*\***

---

Figure 8: Prompt template of targeted modification for instruction-following tasks.

in [1, 2, 3]. Our results indicate that using a batch size of 128 and a single training epoch consistently produces the best outcomes across all methods. Consequently, we adopted these parameters for all subsequent preference optimization experiments. We also configured the maximum sequence length to 2048 and employed a cosine learning rate schedule with a 10% warmup period for training on the preference optimization dataset. For method-specific training hyperparameters, we individually search the learning rates in the range of [3e-7, 5e-7, 6e-7, 1e-6] for each method. Besides, we conduct a grid search according to Table 7 and report the best performance. Table 8 shows the hyperparameters of our method used under each setting.

Table 7: Various preference optimization objectives and hyperparameter search range.

| Method | Objective | Hyperparameter |
|---|---|---|
| FIGA | $-\sum_{\tilde{y}_w^t \in diff(\tilde{y}_w \mid y_l)} \alpha \log \pi_\theta(\tilde{y}_w^t \mid \tilde{y}_w^{<t}, x)$ $+\sum_{y_l^t \in diff(y_l \mid \tilde{y}_w)} \beta \log \pi_\theta(y_l^t \mid y_l^{<t}, x)$ | $\alpha \in [0.5, 1.0, 1.5]$ $\beta \in [0.5, 1.0, 1.5]$ |
| IPO | $\left( \log \frac{\pi_\theta(y_w \mid x)}{\pi_{\text{ref}}(y_w \mid x)} - \log \frac{\pi_\theta(y_l \mid x)}{\pi_{\text{ref}}(y_l \mid x)} - \frac{1}{2\tau} \right)^2$ | $\tau \in [0.01, 0.1, 0.5, 1.0]$ |
| ORPO | $-\log p_\theta(y_w \mid x) - \lambda \log \sigma \left( \log \frac{p_\theta(y_w \mid x)}{1 - p_\theta(y_w \mid x)} - \log \frac{p_\theta(y_l \mid x)}{1 - p_\theta(y_l \mid x)} \right),$ where $p_\theta(y \mid x) = \exp \left( \frac{1}{\mid y \mid} \log \pi_\theta(y \mid x) \right)$ | $\lambda \in [0.1, 0.5, 1.0, 2.0]$ |
| R-DPO | $-\log \sigma \left( \beta \log \frac{\pi_\theta(y_w \mid x)}{\pi_{\text{ref}}(y_w \mid x)} - \beta \log \frac{\pi_\theta(y_l \mid x)}{\pi_{\text{ref}}(y_l \mid x)} - (\alpha \mid y_w \mid - \alpha \mid y_l \mid) \right)$ | $\alpha \in [0.05, 0.1, 0.5, 1.0]$ $\beta \in [0.01, 0.05, 0.1]$ |
| SimPO | $-\log \sigma \left( \frac{\beta}{\mid y_w \mid} \log \pi_\theta(y_w \mid x) - \frac{\beta}{\mid y_l \mid} \log \pi_\theta(y_l \mid x) - \gamma \right)$ | $\beta \in [2.0, 2.5]$ $\gamma \in [0.3, 0.5, 1.0, 1.2, 1.4, 1.6]$ |
| DPO | $-\log \sigma \left( \beta \log \frac{\pi_\theta(y_w \mid x)}{\pi_{\text{ref}}(y_w \mid x)} - \beta \log \frac{\pi_\theta(y_l \mid x)}{\pi_{\text{ref}}(y_l \mid x)} \right)$ | $\beta \in [0.01, 0.05, 0.1]$ |
| DPO-BMC | $\log \sigma \left( \beta \sum_{\tilde{y}_w^t \in \tilde{y}_w} \lambda_{\tilde{y}_w^t} \log \frac{\pi_\theta(\tilde{y}_w^t \mid \tilde{y}_w^{<t}, x)}{\pi_{\text{ref}}(\tilde{y}_w^t \mid \tilde{y}_w^{<t}, x)} \right.$ $\left. -\beta \sum_{y_l^t \in y_l} \lambda_{y_l^t} \log \frac{\pi_\theta(y_l^t \mid y_l^{<t}, x)}{\pi_{\text{ref}}(y_l^t \mid y_l^{<t}, x)} \right),$ where $\lambda_{\tilde{y}_w^t} = \begin{cases} 1 + \min \left( sg \left( \frac{1}{\pi_\theta(\tilde{y}_w^t \mid \tilde{y}_w^{<t}, x)} \right), \delta \right), \text{if } \tilde{y}_w^t \in diff(\tilde{y}_w \mid y_l) \\ 1, \text{otherwise} \end{cases}$ $\lambda_{y_l^t} = \begin{cases} 1 + \min \left( sg \left( \frac{1}{\pi_\theta(y_l^t \mid y_l^{<t}, x)} \right), \delta \right), \text{if } y_l^t \in diff(y_l \mid \tilde{y}_w) \\ 1, \text{otherwise} \end{cases}$ | $\beta \in [0.01, 0.05, 0.1]$ $\delta \in [1.5, 2.0, 2.5, 3.0, 3.5]$ |

Table 8: Hyperparameter values for diverse training settings in DPO-BMC.

| Task | Model | Learning Rate | $\beta$ | $\delta$ |
|---|---|---|---|---|
| QA | Llama2-7B-base | 5e-7 | 0.05 | 3.0 |
| Math | Llama2-7B-base | 5e-7 | 0.05 | 2.5 |
| IF | Llama3-8B-base Mistral-7B-base | 5e-7 5e-7 | 0.01 0.01 | 2.0 2.0 |

# B COST ANALYSIS

## B.1 COST OF BRIDGING PHASE

The Bridging Phase, responsible for synthesizing pseudo-winning responses, operates exclusively **offline**, meaning it incurs no runtime cost during model training. The data synthesis process is designed to be efficient, as it does not require iterative computations or model updates.

For context, we estimated the budget for data synthesis using the `gpt-4-0125-preview` API, based on the API's pricing of \$0.01 per 1K input tokens and \$0.03 per 1K output tokens. Table 9 lists the breakdown of the estimated costs for our three evaluated tasks, which demonstrates that this is a manageable expenditure.

Table 9: Estimated budget for data synthesis using the `gpt-4-0125-preview` API.

| Task | # of Samples | Avg. Input Token Length | Avg. Output Token Length | Cost (\$) |
|---|---|---|---|---|
| QA | 15,732 | 206 | 25 | 44.21 |
| Math | 40,000 | 429 | 47 | 228.00 |
| IF | 61,135 | 728 | 235 | 876.06 |

**Can an Open-Source LLM be Utilized as an Alternative?**   In Table 4, we explore the impact of LLMs on targeted modifications during the Bridging Phase. Our findings indicate that substituting the `gpt-4-0125-preview` model with a less powerful yet open-source alternative, such as `Llama3-70B-Instruct`, **yields comparable performance while significantly surpassing vanilla DPO**. The `Llama3-70B-Instruct` model can be deployed on only 2 NVIDIA-3090 GPUs, with the option to further reduce hardware requirements through low-bit quantization[4]. This provides an economical alternative for our Bridging Phase without compromising performance. Numerous studies have highlighted the superior text modification capabilities of LLMs. For example, LLMs have been effectively employed in synthesizing high-quality data (Wang et al., 2023; Jiang et al., 2023b; 2024a;b; Liu et al., 2024b). Additionally, Ji et al. (2024) show that LLMs can transform initial outputs from upstream models into more helpful and benign responses, thereby aligning generated content with human intentions. In conclusion, our framework demonstrates robustness in leveraging diverse LLMs for targeted modifications, confirming its adaptability and effectiveness.

## B.2   Cost of Modeling Phase

Our Modeling Phase adds minimal computational overhead compared to vanilla DPO. Specifically:

- **Token Difference Identification**: Using a dynamic programming algorithm (edit distance) to identify differing tokens between the pseudo-winning and losing responses. This is a lightweight operation and introduces negligible runtime cost.
- **Reward Weighting Calculation**: We calculate a weighting factor based on the policy model's probability of the identified tokens, which is already computed in the standard DPO setup. Because we halt gradient backpropagation for the weighting factor, this operation does not introduce additional computational costs.

Table 10 demonstrates the comparison of the training times between DPO and DPO-BMC on 4×A800 GPUs, illustrating that **DPO-BMC increases training time by less than 1% across all evaluated tasks**.

Table 10: Runtime usage for DPO and DPO-BMC during the Modeling Phase.

| Task | Base Model | Runtime of DPO (s) | Runtime of DPO-BMC (s) | Increase (%) |
|------|-----------|--------------------|------------------------|--------------|
| QA   | Llama2-7B | 2,831              | 2,850                  | **0.67%**    |
| Math | Llama2-7B | 9,586              | 9,641                  | **0.57%**    |
| IF   | Llama3-8B | 16,179             | 16,318                 | **0.86%**    |

Overall, these results validate that the computational overhead introduced by BMC is minimal, and the approach is highly efficient in terms of runtime, making it practical for real-world applications without significantly increasing resource requirements.

---

[4]https://github.com/ollama/ollama

## C  GRADIENT ANALYSIS

For a mechanistic understanding of our method, we examine the gradients of the loss function $\mathcal{L}_{\text{DPO}}$ in Eq. (2) and $\mathcal{L}_{\text{DPO-BMC}}$ in Eq. (4). Their gradients with respect to the parameters $\theta$ can be written as:

$$
\nabla_\theta \mathcal{L}_{\text{DPO}}(\pi_\theta; \pi_{\text{ref}}) = -\beta \mathbb{E}_{(x,y_w,y_l) \sim \mathcal{D}} \left[ \sigma(\Delta_1) \left[ \underbrace{\nabla_\theta \log \pi_\theta(y_w \mid x)}_{\text{increase likelihood of } y_w} - \underbrace{\nabla_\theta \log \pi_\theta(y_l \mid x)}_{\text{decrease likelihood of } y_l} \right] \right],
$$

where $\Delta_1 = \beta \log \frac{\pi_\theta(y_l|x)}{\pi_{\text{ref}}(y_l|x)} - \beta \log \frac{\pi_\theta(y_w|x)}{\pi_{\text{ref}}(y_w|x)}$.

$$
\nabla_\theta \mathcal{L}_{\text{DPO-BMC}}(\pi_\theta; \pi_{\text{ref}}) = -\beta \mathbb{E}_{(x,\tilde{y}_w,y_l) \sim \tilde{\mathcal{D}}} \left[ \sigma(\Delta_2) \left( \underbrace{\nabla_\theta \log \pi_\theta(\tilde{y}_w \mid x)}_{\text{increase likelihood of } \tilde{y}_w} - \underbrace{\nabla_\theta \log \pi_\theta(y_l \mid x)}_{\text{decrease likelihood of } y_l} \right. \right.
$$

$$
\left. \left. + \underbrace{\sum_{\tilde{y}_w^t \in diff(\tilde{y}_w|y_l)} (\lambda_{\tilde{y}_w^t} - 1)\nabla_\theta \log \pi_\theta(\tilde{y}_w^t \mid \tilde{y}_w^{<t}, x)}_{\text{increase likelihood of desired tokens of } \tilde{y}_w} - \underbrace{\sum_{y_l^t \in diff(y_l|\tilde{y}_w)} (\lambda_{y_l^t} - 1)\nabla_\theta \log \pi_\theta(y_l^t \mid y_l^{<t}, x)}_{\text{decrease likelihood of undesired tokens of } y_l} \right) \right],
$$

where $\Delta_2 = \beta \sum_{y_l^t \in y_l} \lambda_{y_l^t} \log \frac{\pi_\theta(y_l^t|y_l^{<t},x)}{\pi_{\text{ref}}(y_l^t|y_l^{<t},x)} - \beta \sum_{\tilde{y}_w^t \in \tilde{y}_w} \lambda_{\tilde{y}_w^t} \log \frac{\pi_\theta(\tilde{y}_w^t|\tilde{y}_w^{<t},x)}{\pi_{\text{ref}}(\tilde{y}_w^t|\tilde{y}_w^{<t},x)}$.

In contrast to vanilla DPO, which emphasizes sequence-level optimization exclusively, **our proposed method integrates both sequence-level and token-level perspectives**. (1) At the sequence level, we promote preferred completions while penalizing those that are disfavored. (2) At the token level, we further refine the rewards of critical desired and undesired tokens of $\tilde{y}_w$ and $y_l$, respectively. This dual consideration ensures that both the overall sequence structure and the critical token choices are optimized for the desired outcome.

## D  KL DIVERGENCE ANALYSIS DURING TRAINING

In Figure 9, we present the KL divergence between the policy model trained with DPO, DPO-MC, DPO-BC, and DPO-BMC with identical hyperparameters and the reference model, measured on the winning responses from a held-out set of UltraFeedback during training. The results also validate our analyses in §5.2: (1) the Bridging Phase fosters tailored learning toward critical differences in preference data, resulting in more efficient and "sharp" training with a larger KL divergence; (2) our meticulously designed loss function in the Modeling Phase effectively moderates the optimization intensity across diverse training data, thereby achieving a more controlled and steady KL divergence.

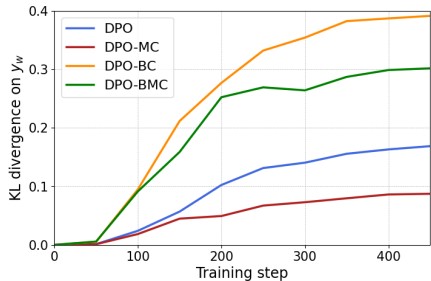

Figure 9: KL divergence from the policy model to the reference model on winning responses of the held-out set of UltraFeedback.

## E  EXPERIMENTS ON LARGER BASE MODELS

We conducted additional experiments using the more capable Qwen2.5-14B-Base model (Team, 2024). As shown in Table 11, our proposed DPO-BMC method delivers even greater performance improvements with this model, achieving a remarkable gain of +8.8 on AlpacaEval 2 and +7.3 on Arena-Hard compared to standard DPO. These results highlight the effectiveness of DPO-BMC scales with model capability, underscoring its potential to deliver even larger gains when applied to more powerful baseline models.

Table 11: Performance comparison across different base models.

| Method | Base Model | LC (%) of AlpacaEval 2 | WR (%) of Arena-Hard |
|--------|-----------|------------------------|----------------------|
| SFT | Mistral-7B-Base | 8.1 | 2.2 |
| DPO | Mistral-7B-Base | 15.1 | 13.6 |
| DPO-BMC | Mistral-7B-Base | **20.8 (+5.7)** | **17.6 (+4.0)** |
| SFT | Llama3-8B-Base | 7.5 | 2.6 |
| DPO | Llama3-8B-Base | 16.0 | 17.6 |
| DPO-BMC | Llama3-8B-Base | **22.4 (+6.4)** | **18.1 (+0.5)** |
| SFT | Qwen2.5-14B-Base | 14.1 | 11.2 |
| DPO | Qwen2.5-14B-Base | 36.3 | 33.6 |
| DPO-BMC | Qwen2.5-14B-Base | **45.1 (+8.8)** | **40.9 (+7.3)** |

## F  CASE STUDY

In this case study, we analyze the performance of various methods, all based on the Llama3-8B-base model, using a sample instruction from AlpacaEval 2. Table 12 displays the responses generated by four different methods. The SFT model produces a concise and factually accurate answer, although it lacks detail and context regarding the name change. In contrast, models further trained with SimPO and DPO yield incorrect responses, asserting that Facebook Corporation did not change its legal name and providing an inaccurate account of the rebranding process. Our proposed method, DPO-BMC, demonstrates a notable improvement by providing a response that is both factually accurate and comprehensive. Additionally, the response generated by our method is significantly shorter than those produced by DPO and SimPO. This combination of detail, accuracy, and brevity highlights the superiority of our approach in delivering informative and precise answers.

Table 12: Generated responses from different methods for a sampled instruction in AlpacaEval 2.

| | |
|---|---|
| **Instruction** | Did Facebook corporation change its name? |
| **SFT** | Yes, Facebook Corporation changed its name to Meta Platforms, Inc. In 2021. |
| **SimPO** | No, the Facebook Corporation did not alter its legal name. Despite the fact that the company has introduced a diverse array of products and services beyond its primary social networking platform, which includes popular applications such as Instagram, WhatsApp, Messenger, and various others, the official name of the corporation continues to be Facebook, Inc. Nevertheless, in October 2021, Facebook made an announcement indicating their intention to modify their corporate brand name to better reflect their expanding range of offerings. As of now, the new brand name has not been disclosed to the public, and it is anticipated that the complete transition to this new name will require a significant amount of time. |
| **DPO** | No, Facebook Corporation did not change its name. Facebook Corporation, also known as Facebook, Inc., is the parent company of the Facebook social media platform, and it has not gone through any significant name change. The company holds other online platforms and services such as Instagram and WhatsApp, but the name Facebook Corporation remains the same. There have been some changes in the branding, such as the rebranding of WhatsApp and Facebook Messenger with the Meta logo, but the parent company's name has not changed. The announcement made by Facebook in October 2021 about rebranding its parent company as Meta did not materialize, and Facebook Corporation still holds the name Facebook Corporation. |
| **DPO-BMC** | Yes, Facebook Corporation changed its name to Meta Platforms, Inc. In October 2021. The rebranding aims to reflect the company's expanded focus beyond social media and into building the "metaverse," a virtual space where people can interact and communicate in immersive, 3D environments. The new name "Meta" represents the company's ambition to drive breakthroughs in areas such as virtual and augmented reality. Despite the name change, Facebook remains the name of the company's flagship social network. |

