# OpenReview forum: "Bridging and Modeling Correlations in Pairwise Data for Direct Preference Optimization"
_ICLR.cc/2025/Conference — ICLR 2025 Poster_

### Official Review · Reviewer_WKKA · 2024-11-02

**Soundness:** 3
**Presentation:** 3
**Contribution:** 2
**Rating:** 6
**Confidence:** 3

**Summary:**

The paper introduces a novel framework called **Bridging and Modeling Correlations (BMC)** to enhance Direct Preference Optimization (DPO) in aligning large language models (LLMs) with human preferences using pairwise preference data. The framework consists of two phases:

1. **Bridging Phase**: Enhances the correlation between winning and losing responses by generating a pseudo-winning response (ỹ_w) through targeted modifications of the losing response (y_l), referencing the winning response (y_w).

2. **Modeling Phase**: Improves token-level credit assignment by dynamically adjusting token rewards based on the policy model's confidence during training. This is formalized in their proposed objective function with adaptable weights.

Despite these new methods, the paper's experimental results show **modest performance improvements** compared to the computational and resource costs involved in the data modification process with stronger LLMs. I would like to understand these aspects further during the rebuttal period.

**Strengths:**

1. **New Approach**: Introducing the Bridging Phase to enhance correlations in preference data is a new solution to a recognized problem in DPO methods.

2. **Dynamic Token-Level Adjustment**: Adjusting token rewards based on model confidence is a novel idea that could potentially improve fine-grained learning.

3. **Comprehensive Evaluation**: The authors conduct extensive experiments across multiple tasks, including question answering, mathematical reasoning, and instruction following.

4. **Versatility**: The framework is adaptable to various DPO variants, demonstrating its potential applicability in different contexts.

**Weaknesses:**

1. **Marginal Performance Improvement vs. Cost**: The reported performance gains, while present, are relatively modest (e.g., 3.8 percentage points improvement in QA tasks), considering the significant effort and computational resources required for data synthesis and modification. Modifying 60-70% of the dataset introduces substantial overhead.

2. **Dependency on Large LLMs for Data Synthesis**: The Bridging Phase relies on powerful LLMs (e.g., GPT-4) for targeted modifications, which may not be feasible or cost-effective for many practitioners.

3. **Insufficient Analysis of Trade-offs**: The paper lacks a detailed discussion on the trade-off between the cost of data modification and the performance gains achieved.

4. **Limited Improvement Over Baselines**: Given the additional complexity and cost, the improvements over standard DPO and other baselines are insufficient to justify the added effort.

**Questions:**

1. **Cost-Benefit Justification**: Can the authors provide a more detailed analysis of the computational and resource costs associated with the Bridging Phase relative to the performance gains? How do the authors justify the added complexity and cost?

2. **Impact of Data Modification Proportion**: Did the authors investigate the relationship between the proportion of data modified in the Bridging Phase and the resulting performance? Is there a point of diminishing returns?

3. **Baseline LLM Limitations**: Is the modest performance improvement partly due to inherent limitations of the baseline LLM used in the authors' experiments? Would using a more capable baseline model lead to greater improvements, or are there fundamental constraints in the proposed approach? It would be beneficial to elaborate on the capabilities of larger baseline models as well, if possible.

---

> ### Author Response · Authors · 2024-11-17
> **Response to Reviewer WKKA (1/2)**
>
> Thank you for your valuable comments! We're glad that you found our work novel, comprehensive, and adaptable. We have listed our response to your concerns as follows.
>
> ---
>
> > W1,3,4 & Q1: Marginal Performance Improvement vs. Cost
>
> We appreciate the reviewer's comment regarding the computational costs and trade-offs. However, we respectfully disagree with the assertion that the performance improvements are marginal when considering the overhead introduced by our method. **The computational cost of our approach is minimal**, and we provide a detailed analysis to demonstrate its efficiency.
>
> i) **Cost of Modeling Phase**
>
> Our Modeling Phase adds minimal computational overhead compared to vanilla DPO. Specifically:
>
> - *Token Difference Identification*: Using a dynamic programming algorithm (edit distance) to identify differing tokens between the pseudo-winning and losing responses. This is a lightweight operation and introduces negligible runtime cost.
> - *Reward Weighting Calculation*: We calculate a weighting factor based on the policy model's probability of the identified tokens, which is already computed in the standard DPO setup. Because we halt gradient backpropagation for the weighting factor, this operation does not introduce additional computational costs.
>
> Below is a comparison of the training times between DPO and DPO-BMC on 4×A800 GPUs, illustrating that **DPO-BMC increases training time by less than 1% across all evaluated tasks**:
>
> | Task | Base Model | Training Time of DPO (s)  | Training Time of DPO-BMC (s) | Increase (%) |
> |:--------------|:--------------:|:---------:|:---------:|:---------:|
> QA|	Llama2-7B|	2,831|	2,850|	**0.67%**
> Math|	Llama2-7B|	9,586|	9,641|	**0.57%**
> IF|	Llama3-8B|	16,179|	16,318|	**0.86%**
>
>
> ii) **Cost of Bridging Phase**
>
> The Bridging Phase, responsible for synthesizing pseudo-winning responses, operates exclusively **offline**, meaning it incurs no runtime cost during model training. The data synthesis process is designed to be efficient, as it does not require iterative computations or model updates.
>
> Moreover, as discussed in lines 292-293 and Appendix B, we demonstrate that using a less powerful, open-source model, such as **Llama3-70B-Instruct, can deliver comparable results to gpt-4-0125-preview**. The Llama3-70B-Instruct model can be deployed on only 2 NVIDIA-3090 GPUs, with the option to further reduce hardware requirements through low-bit quantization [1]. This provides an economical alternative for our Bridging Phase without compromising performance.
>
> For context, we estimated the budget for data synthesis using the gpt-4-0125-preview API, based on the API's pricing of 0.01 per 1K input tokens and 0.03 per 1K output tokens. Below is the breakdown of the estimated costs for our three evaluated tasks, which demonstrates that this is a manageable expenditure:
>
> | Task | # of Samples | Avg. Input Token Length  | Avg. Output Token Length | Cost ($) |
> |:--------------|:--------------:|:---------:|:---------:|:---------:|
> QA|	15,732	|206|	25|	44.21
> Math|	40,000|	429|	47|	228.00
> IF	|61,135	|728	|235	|876.06
>
> iii) **Performance Improvement Justification**
>
> We believe the performance improvement of our method is substantial, not marginal. As shown in Table 2 of our paper, DPO-BMC achieves a further improvement of **3.8 points in QA** and **1.3 points in Math** over DPO, which itself only improves **2.3 and 0.7 points over SFT**, respectively. This incremental gain of DPO-BMC over DPO is not only significant but also practically meaningful, especially given the high baseline performance of DPO.
>
> To further validate the robustness of these improvements, we conducted five independent runs using different random seeds, comparing DPO-BMC to DPO across QA, Math, and instruction following (IF) tasks:
>
> | | Increase on QA | Increase on Math | Increase on IF |
> |:--------------|:-----------:|:--------------:|:---------:|
> Run 1 | 3.8 | 1.3 | 6.4
> Run 2 | 3.7 | 1.4 | 5.9
> Run 3 | 3.9 | 1.3 | 6.3
> Run 4 | 3.6 | 1.2 | 5.8
> Run 5 | 3.7 | 1.4 | 6.1
> **t-statistic** | **73.35** | **35.28** | **53.50**
> **p-value** | **2e-7** | **4e-6** | **7e-7**
>
> These results underscore the consistency and significance of our method's contributions. The high t-statistics and low p-values ($\ll$ 0.05) indicate that the observed improvements are statistically significant, solidifying the reliability of our findings.
>
> Furthermore, in our response to Q3, we demonstrate that DPO-BMC's effectiveness scales with model capability. As more powerful models are adopted, our method is likely to deliver even greater improvements, highlighting its potential for broad applicability and long-term impact.
>
> In conclusion, while our method introduces some computational costs, they are modest and justified by the significant performance improvements achieved.
>
> ### References:
>
> [1] Ollama. https://github.com/ollama/ollama

---

> ### Author Response · Authors · 2024-11-17
> **Response to Reviewer WKKA (2/2)**
>
> > W2: Dependency on Large LLMs for Data Synthesis
>
> As demonstrated in lines 292–293 and detailed in Appendix B, we tested our framework using a less powerful open-source model, Llama3-70B-Instruct, as a substitute for GPT-4 in generating the pseudo-winning responses via targeted modification. Our experimental results in Table 9 (as listed below for your reference) show that **Llama3-70B-Instruct yields performance levels comparable to those achieved with GPT-4**.
>
> | Method | LLM for Targeted Modification | QA | Math  | IF |
> |:--------------|:-----------:|:--------------:|:---------:|:---------:|
> SFT | –  | 56.9  | 47.6  | 7.5
> DPO | –  | 61.3  | 48.3  | 16.0
> DPO-BMC  | Llama3-70B-Instruct  | 64.6  | 49.4  | 21.8
> DPO-BMC  | gpt-4-0125-preview  | **65.1**  | **49.6**  | **22.4**
>
> This finding underscores the adaptability of our method to varying levels of model sophistication, thereby reducing dependence on commercial LLMs without significant impact on final model performance.
>
>
> > Q2: Impact of Data Modification Proportion
>
> To address this, we performed additional experiments across QA, Math, and IF tasks to investigate the relationship between modification proportion and performance gains:
>
> | Data Modification Proportion | QA | Math  | IF |
> |:--------------|:-----------:|:--------------:|:--------------:|
> 100% (DPO-BMC) | **65.1** | **49.6** | **22.4**
> 80% | 64.9 | 49.5 | 22.1
> 60% | 64.5 | 49.3 | 21.5
> 40% | 63.9 | 49.2 | 21.6
> 20% | 63.3 | 49.1 | 19.9
> 0% (DPO-MC) | 62.5 | 48.8 | 17.7
>
> Our results consistently indicate that higher modification proportions yield stronger performance, particularly when increasing from 0% to 20%, which contributes the most significant improvement. This finding underscores the effectiveness of enhancing correlations in pairwise preference data to improve alignment performance.
>
> Our results also show diminishing returns beyond a certain threshold, with 80% modifications achieving comparable results to 100%, as summarized in the table below. This suggests that while modifying a large portion of data is beneficial, it is not strictly necessary to achieve notable gains, thereby offering flexibility in scenarios with computational or data constraints. These findings highlight the scalability and adaptability of our framework, making it practical for diverse applications.
>
> We thank the reviewer for highlighting this point, and we will incorporate this analysis into the revised manuscript to provide a comprehensive view of our method's robustness.
>
>
> > Q3: Baseline LLM Limitations
>
> Thank you for highlighting the potential impact of baseline model capabilities on our results. To investigate this, we conducted further experiments with the more capable **Qwen2.5-14B-Base** model. As shown in the table below, our DPO-BMC method achieves even larger performance gains with this model, with improvements of **+8.8** on AlpacaEval 2 and **+7.3** on Arena-Hard compared to DPO. These results indicate that DPO-BMC's effectiveness scales with model capability, demonstrating its potential to yield even greater improvements with more powerful baselines.
>
> | Method | Base Model | AlpacaEval 2 | Arena-Hard  |
> |:--------------|:-----------:|:--------------:|:---------:|
> SFT | Mistral-7B-Base  | 8.1  | 2.2
> DPO | Mistral-7B-Base  | 15.1  | 13.6
> DPO-BMC  | Mistral-7B-Base  | 20.8 (**+5.7**)  | 17.6 (**+4.0**)
> |-|-|-|-|
> SFT | Llama3-8B-Base  | 7.5  | 2.6
> DPO | Llama3-8B-Base  | 16.0  | 17.6
> DPO-BMC  | Llama3-8B-Base  | 22.4 (**+6.4**) | 18.1 (**+0.5**)
> |-|-|-|-|
> SFT | Qwen2.5-14B-Base  | 14.1  | 11.2
> DPO | Qwen2.5-14B-Base  | 36.3  | 33.6
> DPO-BMC  | Qwen2.5-14B-Base  | 45.1 (**+8.8**)  | 40.9 (**+7.3**)
>
> ---
>
> We sincerely hope our response helps the reviewer better understand how we think about this work. Any future discussion and comments are more than welcome.

---

> > ### Comment · Reviewer_WKKA · 2024-11-23
> >
> > I have carefully reviewed the author's response. While I still have some concerns about the contribution, the response and revised version have helped me better understand the cost-effectiveness of the proposed methodology. Based on this, I am raising my score by one level.

---

> ### Author Response · Authors · 2024-11-24
> **Thanks for your further comments**
>
> Dear Reviewer WKKA,
>
> We sincerely thank you for increasing the score!
>
> Thanks for your constructive review. We greatly appreciate your time and effort in reviewing our paper and reading our comments. Here, We would like to take this opportunity to highlight the key contributions of our work:
> 1. We are the first to identify weak correlations between pairwise data in DPO and address this issue by proposing a novel framework for Bridging and Modeling Correlations (BMC) in pairwise data.
> 2. The Bridging Phase of our framework effectively enhances the informative correlations between pairwise data through targeted modifications. Importantly, this phase is independent of commercial LLMs, ensuring broad applicability.
> 3. We further improve token-level correlation modeling in DPO by dynamically incorporating the policy model's confidence during training, an approach that has not been explored previously.
> 4. Our framework is **effective** (significantly outperforms DPO across 10 datasets), **efficient** (introduces minimal computational overhead), **robust** (validated through extensive quantitative and qualitative analyses), and **versatile** (seamlessly integrates with various DPO variants, consistently boosting their performance).
>
> If you have any additional questions, please feel free to let us know. We are happy to continue addressing any concerns to the best of our ability!
>
> Best regards,
>
> Authors

---

### Official Review · Reviewer_tYGL · 2024-11-03

**Soundness:** 3
**Presentation:** 4
**Contribution:** 3
**Rating:** 6
**Confidence:** 4

**Summary:**

This paper identifies the important role of the correlation between chosen & rejected responses for LLM alignment with DPO-style algorithms. They propose a suite of solutions including (1): rephrasing the chosen response into the style closer to the rejected response to enhance their correlation; (2): adding a token-level weighting in the DPO loss based on the policy model's confidence on the different tokens between the (rephrased) chosen and rejected responses. Extensive experimental results demonstrate that the proposed BMC pipeline could improve the model's performance on QA, math reasoning, and instruction following benchmarks, as well as exhibiting better and more reasonable token-level and sequence-level reward behaviors.

**Strengths:**

* This paper is written with high quality.
* The improvement to the current DPO-style alignment pipeline is well-motivated, with a simple and effective solution.
* The experiment part of this paper is well conducted, showing clear empirical significance with detailed ablation studies.

**Weaknesses:**

* Regarding the "Bridging Phase", I feel it does not necessarily address every aspect of the initial annotations. For example, despite the semantic aspects (i.e., the chosen response is correct while the rejected one is not), the preferred syntax, or linguistic aspects, such as how to better organize the response, might be neglected. If the original chosen response is better in terms of aspects other than correctness, then the Bridging phase may "erase" such useful information.
* The scalability of rephrasing every chosen response is somewhat limited, which can be improved by investigating the effect of executing the BMC for a specific portion of the dataset or considering how to identify the crucial tokens during the DPO training (i.e., based on confidence pattern demonstrated in line 228-245 and Sec 5.3), i.e., designing a better token-level weighting criterion for DPO-MC.
* I think the overall computation overhead introduced by the BMC pipeline should be discussed.

**Questions:**

* Regarding the first point in the weakness part, could the model still be able to learn some preferred linguistic aspect (such as organizing the answer with a CoT style) when the chosen & rejected responses overlap largely in terms of syntax? Does the shared linguistic structure matter or we can achieve a similar effect by identifying the token span that is critical to the semantic (which could be an improved version of DPO-MC with better token weighting criterion)?
* It would be interesting to see why LMs are bad at rephrasing a win response into a lose response. Do you have some observations other than Table 3, such as checking the difference of the result semantic equivalence between $(y_w, \tilde y_w)$ and $(y_l, \tilde y_l)$?
* Do you have some thoughts about why BMC can not boost SimPO?

---

> ### Author Response · Authors · 2024-11-17
> **Response to Reviewer tYGL (1/3)**
>
> Thank you for your insightful and constructive comments! We're glad that you found our work well-motivated, effective, and thorough. We have tried our best to elaborate on the unclear points and revised our paper accordingly.
>
> ---
>
> > W1: Regarding the "Bridging Phase", I feel it does not necessarily address every aspect of the initial annotations. If the original chosen response is better in terms of aspects other than correctness, then the Bridging phase may "erase" such useful information.
>
> We appreciate the reviewer's concern and would like to clarify that our Bridging Phase is carefully designed to preserve the useful aspects of the original winning response while addressing deficiencies in the losing response.
>
> i) **Specific Mechanisms for Preserving Valuable Attributes**
>
> For domain-specific tasks like QA and Math, as evaluated in our paper, syntactic and linguistic features—such as response organization—are predominantly acquired during the supervised fine-tuning (SFT) stage. This ensures that responses adhere to a high standard of linguistic quality even before preference optimization begins. During the Bridging Phase, our focus shifts to addressing the semantic gaps and correctness issues in the losing responses. The targeted modifications are guided by the prompt template illustrated in Figure 7, ensuring that we refine the losing response without altering its core stylistic attributes already established during SFT.
>
> For more general instruction-following tasks, we agree that syntax and linguistic organization play a critical role in user preference. To address this, we account for these aspects during the synthesis of pseudo-winning responses. As shown in the prompt template in Figure 8, the off-the-shelf LLM is instructed to improve the losing response minimally and effectively while using the winning response as a reference:
>
> `"Response 1 is more preferred by humans than Response 2. Please make the minimal necessary changes to Response 2 to improve it, referring to Response 1."`
>
> This process ensures that the improvements focus on both correctness and stylistic quality, retaining any superior syntactic or linguistic features from the original winning response.
>
> ii) **Empirical Evidence**
>
> To further support our claim, we conducted an ablation experiment (refer to W2.1) where the Bridging Phase was selectively applied to a portion of the dataset to preserve certain features of the original winning responses. This experimental setup demonstrated that our full BMC approach, which comprehensively refines the losing responses, outperforms the mixed approach. This confirms that our method harmonizes correctness improvements with linguistic quality retention, addressing the reviewer's concern empirically.
>
> iii) **Future Directions for Enhancement**
>
> We acknowledge that certain stylistic preferences may require more explicit modeling in some cases. As part of future work, we aim to incorporate fine-grained guidance mechanisms, such as external style evaluators or linguistic quality metrics, to further enhance the preservation of stylistic attributes. These extensions could ensure even greater alignment between the synthesized pseudo-winning responses and the full spectrum of desirable features in the original winning response.
>
>
> > W2.1: The scalability of rephrasing every chosen response is somewhat limited, which can be improved by investigating the effect of executing the BMC for a specific portion of the dataset.
>
> To address this, we performed additional experiments across QA, Math, and instruction following (IF) tasks to investigate the relationship between modification proportion and performance gains:
>
> | Data Modification Proportion | QA | Math  | IF |
> |:--------------|:-----------:|:--------------:|:--------------:|
> 100% (DPO-BMC) | **65.1** | **49.6** | **22.4**
> 80% | 64.9 | 49.5 | 22.1
> 60% | 64.5 | 49.3 | 21.5
> 40% | 63.9 | 49.2 | 21.6
> 20% | 63.3 | 49.1 | 19.9
> 0% (DPO-MC) | 62.5 | 48.8 | 17.7
>
> Our results consistently indicate that higher modification proportions yield stronger performance, particularly when increasing from 0% to 20%, which contributes the most significant improvement. This finding underscores the effectiveness of enhancing correlations in pairwise preference data to improve alignment performance.
>
> Our results also show diminishing returns beyond a certain threshold, with 80% modifications achieving comparable results to 100%, as summarized in the table below. This suggests that while modifying a large portion of data is beneficial, it is not strictly necessary to achieve notable gains, thereby offering flexibility in scenarios with computational or data constraints. These findings highlight the scalability and adaptability of our framework, making it practical for diverse applications.
>
> We thank the reviewer for highlighting this point, and we will incorporate this analysis into the revised manuscript to provide a comprehensive view of our method's robustness.

---

> ### Author Response · Authors · 2024-11-17
> **Response to Reviewer tYGL (2/3)**
>
> > W2.2: Consider other ways to identify the crucial tokens during the DPO training, i.e., designing a better token-level weighting criterion for DPO-MC.
>
> We appreciate the reviewer's suggestion to explore alternative token-level weighting criteria for DPO training. To address this, we implemented a method inspired by [1], which leverages an external LLM (GPT-4) to identify crucial tokens and assess their contributions to sentence quality or human preference. Specifically, for a given input, we compared the winning and losing responses token by token, requiring GPT-4 to assign a score to each token based on how much it improves or decreases the overall quality. These scores were normalized and used as token-level weightings in the reward calculation. We refer to this method as "External Weighted."
>
> While this approach represents a reasonable alternative, its performance, as shown in the table below, falls short of our proposed DPO-BMC framework:
>
> | Method | QA | Math  | IF |
> |:--------------|:--------------:|:---------:|:---------:|
> SFT  | 56.9  | 47.6  | 7.5
> DPO | 61.3  | 48.3  | 16.0
> DPO (External Weighted)| 61.5 | 48.2 |  17.2
> DPO-BMC (Ours) | **65.1**  | **49.6**  | **22.4**
>
> The results suggest that while "External Weighted" offers slight improvements over DPO in some scenarios, it is constrained by its reliance on a fixed, external scoring mechanism. In contrast, our DPO-BMC framework dynamically adjusts token-level rewards during training based on the policy model's confidence. This adaptive mechanism allows DPO-BMC to better capture nuanced token-level distinctions, leading to significant performance gains across all tasks.
>
>
> > W3: The overall computation overhead introduced by the BMC pipeline should be discussed.
>
> Here, we provide a detailed analysis of the computation costs associated with our BMC pipeline.
>
> i) **Cost of Modeling Phase**
>
> Our Modeling Phase adds minimal computational overhead compared to vanilla DPO. Specifically:
>
> - *Token Difference Identification*: Using a dynamic programming algorithm (edit distance) to identify differing tokens between the pseudo-winning and losing responses. This is a lightweight operation and introduces negligible runtime cost.
> - *Reward Weighting Calculation*: We calculate a weighting factor based on the policy model's probability of the identified tokens, which is already computed in the standard DPO setup. Because we halt gradient backpropagation for the weighting factor, this operation does not introduce additional computational costs.
>
> Below is a comparison of the training times between DPO and DPO-BMC on 4×A800 GPUs, illustrating that **DPO-BMC increases training time by less than 1% across all evaluated tasks**:
>
> | Task | Base Model | Training Time of DPO (s)  | Training Time of DPO-BMC (s) | Increase (%) |
> |:--------------|:--------------:|:---------:|:---------:|:---------:|
> QA|	Llama2-7B|	2,831|	2,850|	**0.67%**
> Math|	Llama2-7B|	9,586|	9,641|	**0.57%**
> IF|	Llama3-8B|	16,179|	16,318|	**0.86%**
>
> ii) **Cost of Bridging Phase**
>
> The Bridging Phase, responsible for synthesizing pseudo-winning responses, operates exclusively **offline**, meaning it incurs no runtime cost during model training. The data synthesis process is designed to be efficient, as it does not require iterative computations or model updates.
>
> Moreover, as discussed in lines 292-293 and Appendix B, we demonstrate that using a less powerful, open-source model, such as **Llama3-70B-Instruct, can deliver comparable results to gpt-4-0125-preview**. The Llama3-70B-Instruct model can be deployed on only 2 NVIDIA-3090 GPUs, with the option to further reduce hardware requirements through low-bit quantization [2]. This provides an economical alternative for our Bridging Phase without compromising performance.
>
> For context, we estimated the budget for data synthesis using the gpt-4-0125-preview API, based on the API's pricing of 0.01 per 1K input tokens and 0.03 per 1K output tokens. Below is the breakdown of the estimated costs for our three evaluated tasks, which demonstrates that this is a manageable expenditure:
>
> | Task | # of Samples | Avg. Input Token Length  | Avg. Output Token Length | Cost ($) |
> |:--------------|:--------------:|:---------:|:---------:|:---------:|
> QA|	15,732	|206|	25|	44.21
> Math|	40,000|	429|	47|	228.00
> IF	|61,135	|728	|235	|876.06
>
> Overall, these results validate that the computational overhead introduced by BMC is minimal, and the approach is highly efficient in terms of runtime, making it practical for real-world applications without significantly increasing resource requirements. We commit to including a detailed cost analysis in the revised version to further substantiate this claim.
>
> ### Reference:
> [1] RLAIF vs. RLHF: Scaling Reinforcement Learning from Human Feedback with AI Feedback. Lee et al. ICML 2024.
>
> [2] Ollama. https://github.com/ollama/ollama

---

> ### Author Response · Authors · 2024-11-17
> **Response to Reviewer tYGL (3/3)**
>
> > Q1: Could the model still be able to learn some preferred linguistic aspect when the chosen & rejected responses overlap largely in terms of syntax?
>
> When the winning and losing responses overlap significantly in terms of syntax, our framework naturally shifts the optimization focus toward other critical variances, such as semantic nuances or contextual appropriateness. However, as clarified in our response to W1, the preference for syntactic and linguistic organization has been considered in the design of the data synthesis in our Bridging Phase. This ensures that all valuable aspects of the winning response are effectively captured and reinforced during training.
>
>
> > Q2: It would be interesting to see why LMs are bad at rephrasing a win response into a lose response.
>
> Thank you for raising this intriguing question. LMs are predominantly trained to generate high-quality, human-like outputs, with limited exposure to the distribution of low-quality or flawed data. As a result, generating low-quality outputs that mimic the nuanced patterns and errors of losing responses poses a greater challenge for these models.
>
> To further investigate this issue, we conducted an analysis of the semantic equivalence between $(y_w, \tilde{y_w})$ (winning responses and their rephrased counterparts) and $(y_l, \tilde{y_l})$ (losing responses and their rephrased counterparts). Using the sentence embedding model *all-mpnet-base-v2* [3], we computed semantic similarity scores for these pairs. The results showed a high similarity score of **0.88** for $(y_w, \tilde{y_w})$, indicating that pseudo-winning responses closely align with their original counterparts. In contrast, the similarity score for $(y_l, \tilde{y_l})$ was **0.73**, demonstrating that pseudo-losing responses struggle to faithfully preserve the error patterns and nuanced shortcomings present in the original losing responses.
>
> These findings highlight a fundamental limitation in the ability of LLMs to accurately emulate flawed outputs, further justifying our decision to focus on synthesizing pseudo-winning responses ($\tilde{y_w}$) to enhance the informativeness and consistency of preference data.
>
>
> > Q3: Do you have some thoughts about why BMC can not boost SimPO?
>
> In our initial experiments applying BMC to SimPO, we adopted SimPO's optimal hyperparameters without extensive fine-tuning tailored to SimPO-BMC. Despite this, as presented in Table 5 of our paper, SimPO-BMC still demonstrates an improvement over SimPO in QA tasks (+2.5 points) and Math tasks (+0.1 points). However, we acknowledge its slight underperformance in the instruction following (IF) task.
>
> Upon further analysis, we identify that SimPO introduces *a target reward margin* to the Bradley-Terry objective, aiming to encourage a larger margin between the winning and losing responses. In our BMC framework, pseudo-winning responses synthesized during the Bridging Phase inherently alter the reward distribution. This necessitates careful adjustment of the target reward margin to fully leverage the benefits of BMC. Therefore, we conducted additional experiments by fine-tuning the reward margin hyperparameter for SimPO-BMC specifically on the IF task. Our results reveal that with carefully tuned hyperparameters, SimPO-BMC achieves a **0.6**-point improvement over SimPO on the IF task, effectively showcasing the potential of BMC when appropriately calibrated.
>
>
> ### Reference:
>
> [3] https://huggingface.co/sentence-transformers/all-mpnet-base-v2
>
> ---
>
> We hope our response helps the reviewer understand how we think about this work better, and we welcome the reviewer to communicate with us more about it and help us revise it.

---

> > ### Comment · Reviewer_tYGL · 2024-11-23
> > **Thanks**
> >
> > I would like to thank the authors for their detailed responses and I appreciate the extra results. I will keep my current recommendation unchanged.

---

> > > ### Author Response · Authors · 2024-11-23
> > > **Thanks for your further comments**
> > >
> > > Dear Reviewer tYGL,
> > >
> > > Thank you very much for your time and effort!
> > > Your review really helped us greatly in improving our paper, and we are truly grateful for your comments.
> > > We will incorporate the additional explanations and analyses you suggested in the final version.
> > > If you also have any other questions, please feel free to let us know.
> > > We will continue to try our best to answer for you.
> > >
> > > Best,
> > >
> > > Authors

---

### Official Review · Reviewer_pbH6 · 2024-11-04

**Soundness:** 2
**Presentation:** 2
**Contribution:** 2
**Rating:** 6
**Confidence:** 4

**Summary:**

The paper proposes an effective framework for Bridging and Modeling Correlations in pairwise data, named BMC. It leverages a commercial LLM to rewrite the winning response and defines confidence-based token-level weighting to improve the original DPO. The experiments are extensive.

**Strengths:**

- The idea of using commercial LLM to rewrite the winning response for DPO is interesting.

- The experiments are extensive.

**Weaknesses:**

- Regarding the motivation, the authors state that "yw and yl are generated isolatedly, the correlation between yw and yl can be weak during pairwise preference optimization", but this is not usually the case in practice. For example, a common practice is to let an LLM continue writing a prefix of the winning response to get the losing one. I mean, the correlation between these two responses can be easily established during data collection.

- It is not well explained why "this weak correlation poses a challenge, as the winning response yw may not provide sufficiently informative gradients for adjusting the model’s parameters in relation to yl". This argument is too high-level. Can you delve into the details? Despite the results that have proven this point, I still wonder about the reason behind it.

- The token confidence should not be always reliable and then the loss weighting trick hinges on intensive hyper-parameter tuning. This undermines the promise of the method.

- To what extent does the final performance depend on the rewriting quality of the commercial LLM? Can some open-source ones be leveraged for this and what is their performance?

**Questions:**

See above

---

> ### Author Response · Authors · 2024-11-17
> **Response to Reviewer pbH6 (1/3)**
>
> Thank you for your detailed and meaningful comments! We're glad that you found our work interesting and extensive. Please find below a detailed discussion of the points you have raised.
>
> ---
>
> > W1: Regarding the motivation, the authors state that "$y_w$ and $y_l$ are generated isolatedly, the correlation between $y_w$ and $y_l$ can be weak during pairwise preference optimization", but this is not usually the case in practice. For example, a common practice is to let an LLM continue writing a prefix of the winning response to get the losing one. I mean, the correlation between these two responses can be easily established during data collection.
>
> We agree that practices such as generating the losing response by continuing a prefix of the winning response can establish some degree of correlation between the two. However, existing data construction methods, including the "continue writing" strategy,  do not fully address the nuanced challenges that arise in pairwise preference optimization. Specifically:
>
> i) **Superficial vs. Nuanced Correlation**
>
> While generating losing responses from the prefix of winning responses introduces a surface-level correlation, it often lacks the fine-grained and task-specific alignment necessary for effective optimization. For example, such methods may ensure shared context or semantic consistency but fail to emphasize critical distinctions between human-desired outcomes and suboptimal behaviors. Our framework explicitly focuses on enhancing *informative correlations*—those that highlight fine-grained differences in desired and undesired behaviors, as reflected in token-level variations.
>
> ii) **Limitations of the Continue Writing Approach**
>
> Besides, the "continue writing" strategy has inherent limitations:
> - *Restricted Context Access*: It operates without access to the complete winning response, limiting its ability to contrast critical features of superior responses. By contrast, our Bridging Phase leverages the full context of both the winning and losing responses to construct pseudo-winning responses, enhancing informativeness.
> - *Starting Point Ambiguity*: Identifying an appropriate starting point for continuation is challenging and can lead to inconsistencies or artifacts in the generated losing responses.
> - *Unreliable Quality Assurance*: There is no inherent guarantee that the continuation will consistently be of lower quality, which can result in preference pairs that fail to distinguish between desired and undesired behaviors.
>
> iii) **Empirical Evidence**
>
> To further investigate, we implemented the "continue writing" strategy using the following steps:
>
>     a. Splitting the winning response into sentences and randomly selecting a starting point for continuation.
>     b. Using the SFT model to generate a losing response continuation.
>     c. Employing GPT-4 to filter cases where the continuation was not worse than the winning response.
>     d. Training a model with these pairs via DPO.
>
> The table below summarizes the evaluation results across tasks:
>
> | Method | QA | Math  | IF |
> |:--------------|:--------------:|:---------:|:---------:|
> SFT  | 56.9  | 47.6  | 7.5
> DPO | 61.3  | 48.3  | 16.0
> DPO + continue writing | 60.9 | 47.8 |  15.2
> DPO + our data synthesis (DPO-BC) | **63.4** | **49.1** | **20.6**
>
> The results demonstrate that the "continue writing" approach slightly underperforms standard DPO, likely due to the aforementioned limitations. In contrast, our data synthesis method yields significant improvements across all tasks, underscoring the effectiveness of the Bridging Phase in constructing preference pairs that emphasize informative correlations critical for optimizing human-desired behaviors.
>
> We sincerely appreciate your feedback and will include this detailed analysis and corresponding empirical results in our revision.

---

> ### Author Response · Authors · 2024-11-17
> **Response to Reviewer pbH6 (2/3)**
>
> > W2: It is not well explained why "this weak correlation poses a challenge, as the winning response $y_w$ may not provide sufficiently informative gradients for adjusting the model's parameters in relation to $y_l$".
>
> Thank you for raising this point. We appreciate the opportunity to elaborate further. Below, we provide a deeper theoretical and empirical rationale for this issue:
>
> i) **Weak Correlations Lead to Gradient Signal Dilution**
>
> In the context of DPO, the Bradley-Terry objective relies on the relative likelihoods of $y_w$ and $y_l$ to compute gradients. When the correlation between $y_w$ and $y_l$ is weak, the differences between these responses are often superficial (e.g., stylistic or irrelevant variations) rather than substantive distinctions that reflect human-preferred behaviors. Consequently, the optimization process may inadvertently focus on minor discrepancies rather than meaningful distinctions. This results in gradients that are less informative for guiding the model towards robust preference alignment.
>
> ii) **Weak Correlations Lead to Suboptimal Fine-Tuning Focus**
>
> In DPO, the policy model learns to assign higher probabilities to $y_w$ compared to $y_l$. When the pair lacks meaningful correlations, the optimization process tends to distribute rewards uniformly across tokens (as evidenced in Figure 6 of the paper). This uniformity neglects critical nuances, leading to:
>
> - *Overgeneralization*, where the model reinforces broadly correct patterns without learning the fine-grained distinctions that separate superior and inferior responses.
> - *Inefficient learning*, as the model wastes capacity on variations that do not impact alignment with human-desired behaviors.
>
> iii) **The Role of our Bridging Phase in Mitigating These Issues**
>
> By synthesizing the pseudo-winning response $\tilde{y}_w$ using targeted modifications of $y_l$ with $y_w$ as a reference, our framework enhances the correlation between pairwise data. This results in:
>
> - *Sharper Gradients*: The token-level differences between $\tilde{y}_w$ and $y_l$ emphasize critical distinctions while filtering out noise, thereby providing more informative signals during training.
> - *Improved Focus*: The model learns to prioritize nuanced, meaningful distinctions, enabling efficient and robust preference alignment.
>
> We appreciate your feedback and will incorporate this detailed explanation into our revised submission to enhance clarity.

---

> ### Author Response · Authors · 2024-11-17
> **Response to Reviewer pbH6 (3/3)**
>
> > W3: The token confidence should not be always reliable and then the loss weighting trick hinges on intensive hyper-parameter tuning.
>
> We appreciate the reviewer's concern about the reliability of token confidence and its dependency on hyperparameter tuning. We address these points below:
>
> i) **Reliability of Token Confidence & Hyperparameter Robustness**
>
> While token confidence may not always be perfectly reliable, **its utility and effectiveness have been well-documented in prior works**, including high-quality data selection for enhancing language model self-training [1, 2], token-level self-evolution training [3], and as a critical signal for detecting anomalous behaviors [4, 5]. These studies highlight the robustness of token confidence as a valuable learning signal under various scenarios.
>
> To mitigate potential inaccuracies in the policy model's confidence, we incorporate an upper limit threshold $\tau$ (See Eq. (5) and Eq. (6)) to regulate the weight assigned to critical tokens. This mechanism ensures that the model avoids excessively aggressive updates, promoting stability in the learning process. Our ablation study (Figure 3 and lines 403–409) demonstrates the effectiveness of this design, showing that **the model achieves strong performance improvements and stability over a broad range of $\tau$ values**. Importantly, the results highlight that our approach is not overly sensitive to hyperparameter tuning, as performance remains robust within reasonable thresholds of $\tau$.
>
> ii) **Comparison with External Token-Level Weighting**
>
> To further substantiate our approach, we compared our **dynamic token-level weighting mechanism** to an **external weighting method** inspired by [6]. In this baseline, we leveraged GPT-4 to assign token-level scores based on their contribution to response quality. These scores were normalized and used as weights in the reward calculation. The table below summarizes the results:
>
> | Method | QA | Math  | IF |
> |:--------------|:--------------:|:---------:|:---------:|
> SFT  | 56.9  | 47.6  | 7.5
> DPO | 61.3  | 48.3  | 16.0
> DPO (External Weighting)| 61.5 | 48.2 |  17.2
> DPO-BMC (Ours) | **65.1**  | **49.6**  | **22.4**
>
> Although the external weighting method provided slight improvements over DPO, its reliance on fixed external scoring mechanisms limited its adaptability to nuanced token-level distinctions. In contrast, our dynamic weighting mechanism leverages the policy model's evolving confidence during training, enabling context-sensitive adjustments to token-level rewards. This adaptability ensures that the model can emphasize challenging distinctions and reinforce learned patterns dynamically, leading to consistently superior performance across tasks.
>
> **Summary**
>
> Our approach integrates the well-documented benefits of token confidence with safeguards like the upper limit threshold $\tau$, effectively addressing potential variability. Moreover, the comparison with external weighting methods underscores the advantage of our dynamic, context-aware design. We believe these innovations make our method both practical and impactful, significantly advancing state-of-the-art preference optimization techniques.
>
> > W4: To what extent does the final performance depend on the rewriting quality of the commercial LLM? Can some open-source ones be leveraged for this and what is their performance?
>
> As demonstrated in lines 292–293 and detailed in Appendix B, we tested our framework using a less powerful open-source model, Llama3-70B-Instruct, as a substitute for GPT-4 in generating the pseudo-winning responses via targeted modification. Our experimental results in Table 9 (as listed below for reference) show that **Llama3-70B-Instruct yields performance levels comparable to those achieved with GPT-4**.
>
> | Method | LLM for Targeted Modification | QA | Math  | IF |
> |:--------------|:-----------:|:--------------:|:---------:|:---------:|
> SFT | –  | 56.9  | 47.6  | 7.5
> DPO | –  | 61.3  | 48.3  | 16.0
> DPO-BMC  | Llama3-70B-Instruct  | 64.6  | 49.4  | 21.8
> DPO-BMC  | gpt-4-0125-preview  | **65.1**  | **49.6**  | **22.4**
>
> This finding underscores the adaptability of our method to varying levels of model sophistication, thereby reducing dependence on commercial LLMs without significant impact on final model performance.
>
> ### References:
>
> [1] Uncertainty-aware Self-training for Text Classification with Few Labels. Mukherjee et al. NeurIPS 2020.
>
> [2] Uncertainty-Aware Curriculum Learning for Neural Machine Translation. Zhou et al. ACL 2020.
>
> [3] Token-Level Self-Evolution Training for Sequence-to-Sequence Learning. Peng et al. ACL 2023.
>
> [4] On Hallucination and Predictive Uncertainty in Conditional Language Generation. Xiao and Wang. EACL 2021.
>
> [5] Fact-Checking the Output of Large Language Models via Token-Level Uncertainty Quantification. Fadeeva et al. ACL Findings 2024.
>
> [6] RLAIF vs. RLHF: Scaling Reinforcement Learning from Human Feedback with AI Feedback. Lee et al. ICML 2024.

---

> ### Author Response · Authors · 2024-11-24
> **Kindly Reminder for the Discussion**
>
> Dear Reviewer pbH6,
>
> Thanks for your careful reading of our paper! We have tried our best to elaborate on the unclear points and revised our paper accordingly.
>
> 1. We have illustrated the key differences between existing data construction methods and our Bridging Phase. Additionally, we conducted targeted experiments to demonstrate that the "continue writing" strategy you suggested consistently underperforms compared to our proposed approach (addressing Weakness 1).
> 2. Following your insightful suggestions, we expanded our discussion on why weak correlations between pairwise data pose challenges to preference optimization (addressing Weakness 2).
> 3. To address your concerns, we provided comprehensive evidence regarding the reliability of token confidence and the sensitivity to hyperparameters. We also demonstrated the superiority of our dynamic weighting mechanism over static, external methods (addressing Weakness 3).
> 4. We moved Appendix B to Section 5.1 to emphasize that a less powerful open-source model, Llama3-70B-Instruct, can substitute GPT-4 in the Bridging Phase, showcasing the adaptability and cost-effectiveness of our method (addressing Weakness 4).
>
> We would like to know whether you find our response satisfactory, or if there are more questions that we could clarify. Since the public discussion phase will be ending in a few days on November 26th, we are more than happy to hear your comments and address any of your further concerns during the remaining time.
>
> Best regards,
>
> Authors

---

> > ### Comment · Reviewer_pbH6 · 2024-11-25
> > **Reply**
> >
> > I appreciate the effort made by the authors during the rebuttal period and would deliver a weak acceptance suggestion on this submission.

---

> ### Author Response · Authors · 2024-11-25
> **Thanks for your further comments**
>
> Dear Reviewer pbH6,
>
> We sincerely thank you for increasing the score!
>
> Your feedback has been incredibly valuable in enhancing our paper, and we are deeply appreciative of your time and effort. We will incorporate the additional explanations and analyses you suggested in the final version. If you still have any other questions in the future, please feel free to let us know. We will continue to try our best to answer for you.
>
> Best regards,
>
> Authors

---

### Author Response · Authors · 2024-11-21
**General Response to Reviewers and Revision Submitted**

We sincerely thank all the reviewers for their insightful and constructive comments, which have greatly contributed to improving our work. We have thoroughly revised the paper to address the reviewers' concerns. Below we summarize the major revisions (the main revisions are marked with *blue* text in the pdf, we also made some minor layout changes to fit the page limit), while we reply to the comments of each reviewer separately.

**Summary of Major Revisions**

1. We have supplemented a detailed cost analysis of the proposed BMC pipeline in Appendix B, which illustrates that the computational overhead introduced by our approach is minimal (Reviewer tYGL, WKKA).
2. We have relocated Appendix B from our initial manuscript to Section 5.1, emphasizing that a less powerful open-source model, Llama3-70B-Instruct, can serve as a substitute for GPT-4 in the Bridging Phase. This revision highlights the adaptability and cost-effectiveness of our approach (Reviewer pbH6, WKKA).
3. We have investigated the relationship between the proportion of data modified in the Bridging Phase and the resulting performance (Reviewer tYGL, WKKA).
4. We have expanded the explanation of why weak correlation between pairwise data poses a challenge to preference optimization (Reviewer pbH6), and why LLMs are bad at rephrasing a winning response into a losing response (Reviewer tYGL).
5. We have added more baselines in Section 5.1, including i) continue writing a prefix of the winning response to generate the losing one (Reviewer pbH6) and ii) leveraging an off-the-shelf LLM for external weighting of token-level reward (Reviewer tYGL). These additions provide a deeper understanding of the effectiveness of our method.
6. We have reported additional strong experimental results of our approach based on Qwen2.5-14B-Base in Appendix E, indicating that our method's effectiveness scales with model capability (Reviewer WKKA).

**Conclusion**

We deeply appreciate the reviewers' valuable feedback, which has enabled us to refine our work significantly. These revisions strengthen both the theoretical and practical contributions of our paper. We believe that the additional analyses, new baselines, and expanded experiments provide a comprehensive response to all concerns, making the work stronger and more impactful.

We welcome further questions or feedback and are committed to providing any additional clarifications or materials necessary to assist in your evaluation.

---

### Author Response · Authors · 2024-12-04
**Thanks to all Reviewers and Rebuttal Summary**

Dear Reviewers and ACs,

We sincerely thank all the reviewers for their thoughtful feedback and constructive discussions! Below, we summarize our key contributions, responses to reviewer comments, and the outcome of our rebuttal process.

**Key Contributions**

Our submission proposes BMC, a novel framework for Bridging and Modeling Correlations in pairwise data, enabling more effective Direct Preference Optimization (DPO). Key contributions include:
1. We identify and address weak correlations in pairwise data with a unique framework for Bridging and Modeling Correlations.
2. The Bridging Phase of our framework effectively enhances the informative correlations between pairwise data through targeted modifications. Importantly, this phase is independent of commercial LLMs, ensuring broad applicability.
3. We further improve token-level correlation modeling in DPO by dynamically incorporating the policy model's confidence during training, an approach that has not been explored previously.
4. Our framework is **effective** (significantly outperforms DPO across 10 datasets), **efficient** (introduces minimal computational overhead), **robust** (validated through extensive quantitative and qualitative analyses), and **versatile** (seamlessly integrates with various DPO variants, consistently boosting their performance).

The novelty and soundness of our contributions have been recognized by reviewers as:
- "*interesting*", "*experiments are extensive*" (pbH6)
- "*well-motivated*", "*clear empirical significance with detailed ablation studies*" (tYGL)
- "*novel idea*", "*comprehensive evaluation*" (WKKA)

**Reviewer Comments and Our Responses**

We have thoroughly addressed the reviewer concerns, as summarized below:

| Comment | Response | Revision | Related Reviewers|
|:--------------|:--------------:|:---------:|:---------:|
| Computation cost of BMC should be discussed | Added a detailed cost analysis showing that BMC introduces minimal computational overhead | Appendix B | tYGL (W3); WKKA (W1, W3, W4, Q1) |
| Can open-source LLMs work for the Bridging Phase? | Highlighted Llama3-70B-Instruct as a viable substitute for GPT-4 | L.418-423 | pbH6 (W4); WKKA (W2) |
| Impact of data modification proportion in the Bridging Phase? | Added analysis showing higher modification proportions yield stronger performance | L.412-417 | tYGL (W2.1); WKKA (Q2) |
| Clarify motivation | Expanded explanation of weak correlation issues and challenges | L.150-155 | pbH6 (W1, W2) |
| Why are LLMs bad at rephrasing a winning response into a losing one? | Added qualitative and quantitative analyses | L.407-410 | tYGL (Q2) |
| Consider alternative methods for identifying crucial tokens in DPO | Added a competitive baseline to validate BMC's superiority | L.310-315 | tYGL (W2.2) |
| Does BMC scale with more capable models? | Added experiments with Qwen2.5-14B, showing BMC's effectiveness scales with model capability | Appendix E | WKKA (Q3) |

**Outcome of Rebuttal**

Our responses addressed all reviewer concerns, with no further questions raised. Notably, both Reviewer pbH6 and Reviewer WKKA increased their ratings from 5 to 6.

We deeply appreciate the reviewers' valuable insights, which has significantly enhanced our work. The new analyses, baselines, and experiments strengthen both the theoretical and practical contributions of our paper. We are confident that these revisions make our work more impactful and compelling. We will continue striving for excellence.

Thank you once again for your time and effort!

Best regards,

Authors

---

### Meta-Review · Area_Chair_tvFp · 2024-12-17

**Metareview:**

This paper presents a new framework for training models using DPO.  The core idea is to leverage large language models (LLMs) to refine responses, specifically by increasing the correlation between "winning" (preferred) and "losing" (less preferred) responses. This process enhances the training data and ultimately leads to improved model performance.  The authors demonstrate consistent improvements across various tasks.

Strengths:
- The proposed method is novel and the use of LLMs to manipulate response correlations is an innovative approach that yields performance gains.
- Additional results provided during the rebuttal phase, including experiments with different LLMs, showcase the generalizability of the method.

Weaknesses:
- While the LLM-based response regeneration can be performed offline, it still incurs a significant computational cost (over $200 for some datasets).
- The observed performance improvements are somewhat moderate.

Despite the identified weaknesses, the novelty and effectiveness of the proposed method warrant an acceptance recommendation.

**Additional Comments On Reviewer Discussion:**

The authors have been responsive to the reviewers' concerns, providing additional results during the rebuttal period to strengthen their claims. Notably, they've expanded their experiments to include other LLMs, demonstrating the generalizability of their method, and clarified the novel aspects of their work.

To further enhance the revised manuscript, I recommend addressing the remaining concerns raised by the reviewers, specifically: (i) Computational Complexity: Provide a more detailed analysis of the computational costs involved, both during the offline (training) and online (inference) stages. (ii) Performance Improvement Margins: Discuss the magnitude of the performance gains achieved by the proposed method, acknowledging any limitations or scenarios where the improvements are less pronounced.

---

### Decision · Program_Chairs · 2025-01-22

Accept (Poster)